# Tuning oxidant and antioxidant activities of ceria by anchoring copper single-site for antibacterial application

Peng Jiang[1,4], Ludan Zhang[2,4], Xiaolong Liu[3,4], Chenliang Ye[1], Peng Zhu[1], Ting Tan [3] ✉, Dingsheng Wang [1] ✉ & Yuguang Wang[2] ✉

The reaction system of hydrogen peroxide ($H_2O_2$) catalyzed by nanozyme has a broad prospect in antibacterial treatment. However, the complex catalytic activities of nanozymes lead to multiple pathways reacting in parallel, causing uncertain antibacterial results. New approach to effectively regulate the multiple catalytic activities of nanozyme is in urgent need. Herein, Cu single site is modified on nanoceria with various catalytic activities, such as peroxidase-like activity (POD) and hydroxyl radical antioxidant capacity (HORAC). Benefiting from the interaction between coordinated Cu and $CeO_2$ substrate, POD is enhanced while HORAC is inhibited, which is further confirmed by density functional theory (DFT) calculations. $Cu\text{-}CeO_2 + H_2O_2$ system shows good antibacterial properties both in vitro and in vivo. In this work, the strategy based on the interaction between coordinated metal and carrier provides a general clue for optimizing the complex activities of nanozymes.

In the past decades, the irrational use of antibiotics worldwide has led to the growing problem of bacterial resistance, which has become one of the major threats to public health[1]. Therefore, it is urgent to develop new broad-spectrum antibacterial methods. Recently, the antibacterial application of nanozymes, a new generation of nanomaterials mimicking the catalytic activities of natural enzyme (peroxidase, catalase, oxidase, etc.), has arouse researchers' great concern[2,3]. Among them, some nanozymes are able to catalyze the decomposition of hydrogen peroxide ($H_2O_2$) by simulating the activity of natural peroxidase (POD) to produce highly toxic hydroxyl radicals (·OH), so as to achieve bactericidal effect[4,5]. Of all the massive antibacterial nanozyme families, single site enzymes (SSEs) have been exerting high catalytic POD activities, with maximized atom utilization efficiency[6,7]. Since the substrates such as three-dimensional carbon materials exhibit poor catalytic performance, currently reported nanozyme + $H_2O_2$ system based on atomic level manipulation are mostly focused on modifying metal active sites to obtain POD-like activity[8,9]. Yet, the complex catalytic activities of nanozymes in nanozymes + $H_2O_2$ system should be paid close attention.

As an emerging nanozyme, nanoceria ($CeO_2$) possesses multiple catalytic properties such as peroxidase-like activity (POD), catalase-like activity (CAT), oxidase-like activity (OXD), superoxide dismutase-like activity (SOD) and hydroxyl radical antioxidant activity (HORAC)[10–13]. When these sophisticated catalytic reaction process coexist, their reaction pathways and outcomes may be antagonistic or competitive with others, resulting in negative antibacterial effect which can be even worse than that of applying $H_2O_2$ alone[14]. In other words, ·OH, superoxide anions ($O_2^{·-}$) and $H_2O_2$, the main toxic by-products of aerobic metabolism, can be scavenged by $CeO_2$ when employed as a bio-antioxidant[15,16]. Obviously, as a peroxidase-mimicking nanozyme, this characteristic is not conducive to the forming of reactive oxygen species (ROS) in the presence of $H_2O_2$, thus producing a poor

[1]Department of Chemistry, Tsinghua University, Beijing 100084, China. [2]Center of Digital Dentistry/Department of Prosthodontics, National Center of Stomatology, National Clinical Research Center for Oral Diseases, National Engineering Laboratory for Digital and Material Technology of Stomatology, Beijing Key Laboratory of Digital Stomatology, NHC Research Center of Engineering and Technology for Computerized Dentistry, Peking University School and Hospital of Stomatology, Beijing 100081, China. [3]Laboratory of Theoretical and Computational Nanoscience, CAS Center for Excellence in Nanoscience, National Center for Nanoscience and Technology, Chinese Academy of Sciences, Beijing 100190, China. [4]These authors contributed equally: Peng Jiang, Ludan Zhang, Xiaolong Liu. ✉e-mail: tant@nanoctr.cn; wangdingsheng@mail.tsinghua.edu.cn; wangyuguang@bjmu.edu.cn

physiological activity related to ·OH. Therefore, effective regulation of multiple catalytic properties of $CeO_2$ and their reaction process towards a favorable direction is key to optimizing $CeO_2 + H_2O_2$ antibacterial system.

In addition, extensive investigations have been made focusing on the activity of the metal site itself. Meanwhile, interaction between the metal atoms and the substrate with complex intrinsic catalytic activities is of great significance for regulating the nanozyme activities. Both should be carefully considered while investigate the materials with multi-channel activities. $CeO_2$ has a fluorite-like cubic structure which close-packed cerium atoms are coordinated with eight $O^{2-}$ ions. It possesses a host of oxygen vacancies on its surface in response to the unique shuttle between $Ce^{3+}$ and $Ce^{4+}$ redox states[17]. When it comes to SSEs with 100% atomic efficiency[18–20], metal-support interaction (MSI) plays a pivotal role in modulating electronic structure on the active site, which favors adsorption and desorption of reactive intermediates[21,22]. Such features indicate that its overall performance can be manipulated via tuning the electronic structure of both foreign metal atom and the surrounded Ce atoms. It was reported by Wang[23] that $Pt_1/CeO_2$ catalyst with an asymmetric $Pt_1O_4$ configuration displayed exceptional CO oxidation performance relative to square-planar counterpart owing to the tailoring of the local environment of isolated $Pt^{2+}$. Hensen[24] illustrated that the high mobility of surface lattice oxygen and oxygen atoms spilled over from $Pd-CeO_2$ interface which originated from strong MSI contributed to the high stability of oxidized Pd single atoms during CO oxidation. Li also reported that the incorporation of Mn can boost the catalytic performance of the surrounded Ce atoms[25]. Thus, the incorporation of foreign metals into parent $CeO_2$ will engineer its surface structure and regulate reaction intermediates, resulting an optimum antibacterial performance.

Copper agent, as a long-standing antibacterial agent, has achieved excellent antibacterial effects through electrostatic adsorption, ion permeation, and disruption of bacterial redox homeostasis[26,27]. In recent years, it has attracted widespread attention in simulating natural oxidase and peroxidase for antibacterial purposes[28,29]. However, traditional Cu antibacterial agents often possess a high content of Cu, which not only causes waste of Cu catalytic sites in the core, but also poses a risk of causing damage to normal cells after precipitation in the form of Cu ions in practical applications[30]. The above factors encourage us to seek safer and more efficient Cu antibacterial materials to solve the problems of poor utilization and stability of traditional Cu antibacterial agents. Motivated by the above research findings, we hypothesize that the introduction of Cu single-site into $CeO_2$ can not only effectively regulate the complex catalytic activities of $CeO_2$, but also make up for the low utilization of Cu in traditional Cu antibacterial agents while retaining its antibacterial property.

In this work, we deploy a facile strategy to prepare a Cu single site modified $CeO_2$ nanozyme ($Cu-CeO_2$) by employing nano-$CeO_2$ with multiple catalytic activities as the substrate. Benefiting from the modulation of the reaction energy of potential determining step (PDS) of Ce cite by Cu species, $Cu-CeO_2$ demonstrated an increase in POD-like activity and a decrease in HORAC activity compared to pristine $CeO_2$. In vitro and in vivo tests demonstrate that $CeO_2$ can significantly weaken the inherent antibacterial activity of $H_2O_2$ against Methicillin-resistant *Staphylococcus aureus* (MRSA) and *Escherichia coli* (*E. coli*), which can be effectively reversed by $Cu-CeO_2$. The above findings indicate that Cu single-site modification can effectively regulate the complex catalytic activities of $CeO_2$ nanozyme, and $Cu-CeO_2$ single-site nanozyme has a good prospect in the treatment of drug-resistant bacterial infections.

## Results

### Synthesis and characterization of $Cu-CeO_2$ SSE

The fabrication of $Cu-CeO_2$ catalyst is illustrated in Fig. 1a. First, $CeO_2$ nanosphere is obtained by a solvothermal approach, followed by calcinating step under air condition. Then, Cu ions could be adsorbed and deposited on ceria substrates in the presence of alkaline solutions. Finally, the obtained composite is heated in air first, and then pyrolyzed under hydrogen atmosphere to get the target $Cu-CeO_2$ product. Owing to the Cu-O-Ce interaction, Cu species are present as a dispersive state in $Cu-CeO_2$ sample after calcined in air. The scanning electron microscopy (SEM) image in Fig. 1b and transmission electron microscopy (TEM) image in Fig. 1c clearly reveals that the as-prepared $Cu-CeO_2$ possesses a spherical morphology with a quite rough surface, which is similar to their parent ones in Supplementary Figs. 1, 2. Supplementary Fig. 3 shows high-resolution TEM (HRTEM) image of $Cu-CeO_2$ precursor and the positions circled by two rectangular frames are enlarged on the right. Obviously, two representative lattice fringes with a spacing of 0.28 nm are at an angle of 90° in the upper right. The interplanar spacing of the exposed plane with an included angle of 45° is 0.19 nm. As shown in the lower right of Supplementary Fig. 3, the lattice fringes with an interplanar spacing of 0.31 nm correspond to the (111) plane of the sample. Element mapping results in Supplementary Fig. 4 demonstrate the homogeneous distribution of Cu, O and Ce components throughout the sample. The aberration corrected high angle annular dark field-scanning transmission electron microscopy (AC-HAADF-STEM) result in Supplementary Fig. 5 also shows that there are no obvious crystallized copper species in the form of clusters or small particles in the structure. Energy dispersive spectroscopic (EDS) mapping analysis (Fig. 1d) signifies a uniform distribution of Cu component in $Cu-CeO_2$ catalyst. Figure 1e shows the HRTEM image of $Cu-CeO_2$. It can be seen that there are no clusters or small particles throughout the structure. The results in Fig. 1f, g indicate that the main interplanar spacing is about 3.2 Å by Z-contrast analysis.

Figure 2a shows the X-ray diffraction (XRD) patterns of $CeO_2$ and $Cu-CeO_2$ samples, respectively. It can be seen that the main diffraction peaks of two samples are indexed to the cubic structure of $CeO_2$ (JCPDS card no. 34-0394). Except for the diffraction peaks of $CeO_2$, no agglomerated Cu species are detected, indicating that the structure of $CeO_2$ is not changed after the introduction of Cu. In addition, the XRD pattern of $Cu-CeO_2$ sample is fitted in Supplementary Fig. 6, which yields a crystallite size of smaller particle as 9 nm and the unit cell parameters as $a = b = c = 5.3912$ Å and $\alpha = \beta = \gamma = 90°$. On this basis, the model structural diagram of parent cubic $CeO_2$ is constructed and illustrated in Supplementary Fig. 7. From the visual point of view in Supplementary Fig. 7b, the lattice fringes which correspond to (200) and (002) facets are perpendicular to each other and the lattice fringes at an included angle of 45° belong to the (220) facet, which is consistent with the above results obtained from the electron microscope analysis. In order to further accurately determine the nanostructure of the material, we also carry out Raman spectroscopy measurements. In Fig. 2b, the peak centered at 460 cm$^{-1}$ is attributed to the $F_{2g}$ vibrational mode of $CeO_2$ crystal[31]. Compared with parent $CeO_2$, the softening of the $F_{2g}$ mode of $Cu-CeO_2$ is accompanied by the appearance of a broad feature centered around 600 cm$^{-1}$, which belongs to the defect-induced mode (D)[32]. The appearance of D band is due to the formation of defect species in Ce-O coordination, which leads a consequence that the vibration signal of Ce-O cannot be cancelled in all directions[33]. Since there exist a Cu-O-Ce coordination structure in $Cu-CeO_2$ catalyst, the interaction between Cu-O and Ce-O is not equal, which leads to the difference in the vibration of Ce-O in different directions. Thus, the increasing disordering level of $Cu-CeO_2$ leads to the variation of D band. In addition, Raman peaks at 290 cm$^{-1}$ and 340 cm$^{-1}$, which assigned to CuO phase, are not observed[34,35]. The elemental composition and valence states are investigated by X-ray photoelectron spectroscopy (XPS) technique (Supplementary Fig. 8, Fig. 2c). The XPS spectra of Ce 3d (Fig. 2c) are fitted into 10 peaks, which are attributed to the $Ce^{4+}$ species at v, v'', v''', u, u'', u''' and the $Ce^{3+}$ species at $v_0$, v', $u_0$, u', respectively[23,36–38].

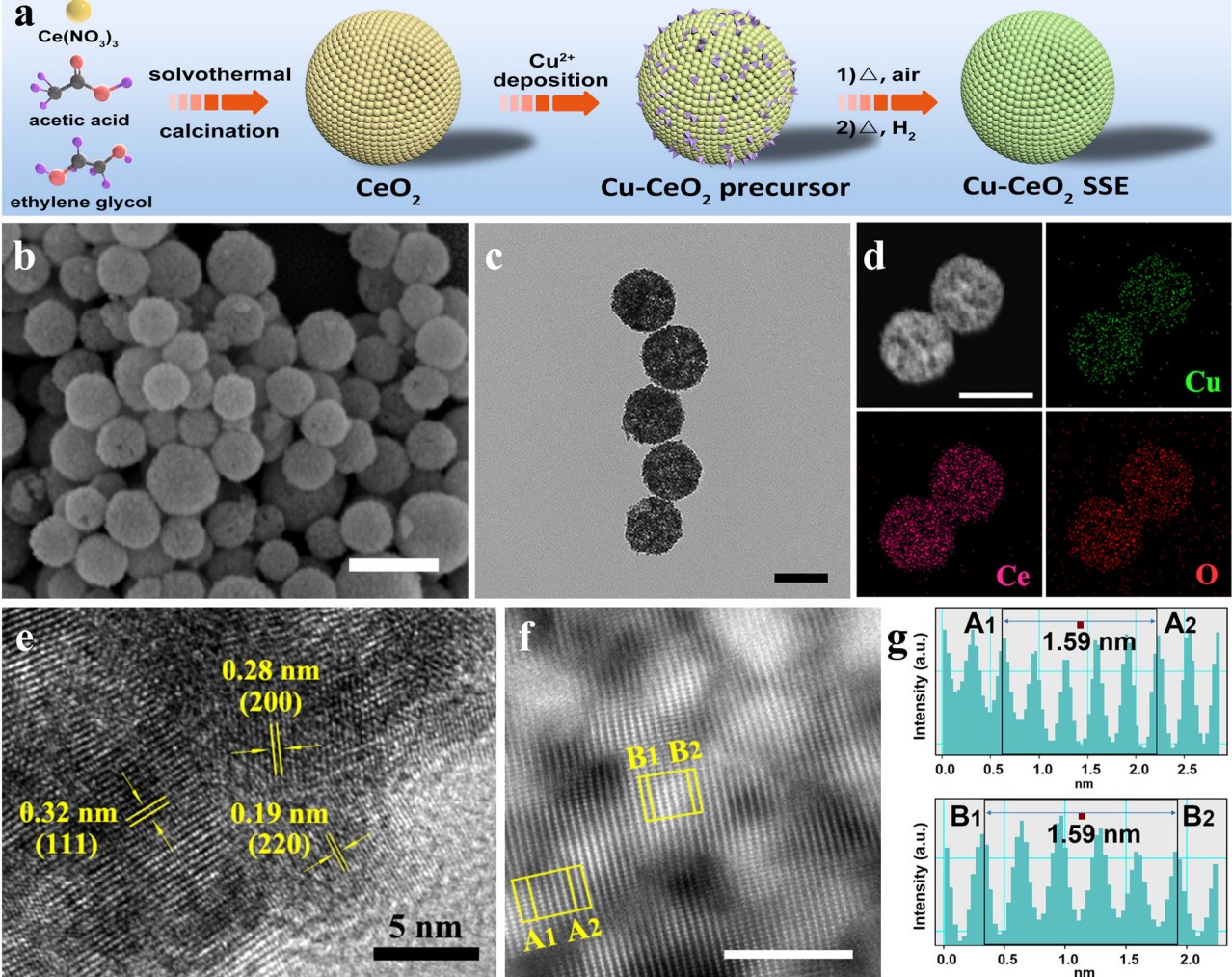

**Fig. 1 | Schematic illustration and morphology characterization for the Cu-CeO₂ catalyst. a** Schematic illustration of the synthetic procedure of Cu-CeO₂ sample, **b** Scanning electron microscopy images of Cu-CeO₂ catalyst (scale bar: 200 nm), **c** Representative TEM image of Cu-CeO₂ catalyst (scale bar: 100 nm), **d** TEM image and corresponding elemental mapping of Cu, O and Ce recorded on the nanoparticles (scale bar: 100 nm), **e** HRTEM images of Cu-CeO₂ catalyst, **f** AC-HAADF-STEM image of Cu-CeO₂ catalyst (scale bar: 5 nm), **g** Corresponding Z-contrast analysis of region A and B in (**f**). All experiments were independently repeated three times with similar results.

Supplementary Table 1 shows the actual $Ce^{3+}/Ce$ ratios of both $CeO_2$ and $Cu\text{-}CeO_2$ samples. It is obvious that the $Ce^{3+}$ content for $Cu\text{-}CeO_2$ is nearly 18% from XPS analysis, which is lower than that of $CeO_2$ (23%). These results indicate that the introduction of Cu element into parent $CeO_2$ would reduce its surface $Ce^{3+}$ concentration. Similar results have also been reported in previous report[38]. According to the authors' calculations and structural characterization, the substitution of one $Ce^{3+}$ adjacent to an oxygen vacancy ($V_O$) by one $Cu^{2+}$ normally accompanies a phenomenon that the other $Ce^{3+}$ would be readily converted to $Ce^{4+}$ for the charge balance, and therefore the introduction of $Cu^{2+}$ will reduce the $Ce^{3+}/Ce^{4+}$ ratio compared with original $CeO_2$. The O 1s spectra of $CeO_2$ and $Cu\text{-}CeO_2$ samples are illustrated in Supplementary Fig. 8b. The peak appearing at approximately 529 eV ($O_I$) can be attributed to the lattice oxygen of $Ce^{4+}$ and the feature at 530.7 eV ($O_{II}$) corresponds to oxygen vacancies or the lower-coordination lattice oxygen of $Ce^{3+}$[36,37]. Also, the higher binding energy feature at 531.6 eV is also associated with the presence of surface hydroxy-containing groups. It is obvious that parent $CeO_2$ possesses a larger $O_{II}/(O_I + O_{II})$ ratio (23.3%) than that of $Cu\text{-}CeO_2$ (19.8%), which is in line with the Ce 3d XPS results.

Considering that the surface oxygen is prone to be removed under vacuum conditions, the structural information of Cu can be better reflected by means of Synchrotron radiation characterization. X-ray absorption fine structure (XAFS) measurements can effectively detect the local coordination states and electronic structure of copper on $CeO_2$ support. Figure 2d shows the Cu K edge X-ray absorption near-edge structure (XANES) profiles of $Cu\text{-}CeO_2$, Cu foil, and CuO samples. According to the position of near-edge absorption energy, it can be concluded that the Cu species bear an oxidation valence state in $Cu\text{-}CeO_2$ sample. The Fourier-transformed (FT) extended X-ray absorption fine structure (EXAFS) curve of $Cu\text{-}CeO_2$ (Fig. 2e) exhibits only one prominent peak at approximately 1.9 Å, corresponding to the first shell of Cu-O scattering interaction. No appreciable Cu-Cu coordination characteristic peak is detected at 2.24 Å, signifying that there is no formation of Cu-Cu bond. We also performed CO-probe molecule Fourier transform infrared (FTIR) measurements to investigate the nature of Cu metal sites. In situ diffused reflectance infrared Fourier transform spectroscopy (DRIFTS) was used to investigate the adsorption ability of $CeO_2$ and $Cu\text{-}CeO_2$. As shown in Supplementary Fig. 9a, $Cu\text{-}CeO_2$ catalyst exhibits a band around 2105 cm$^{-1}$, assigned to the linear CO adsorbed on $Cu^+$ sites ($Cu^+$-CO), indicating that CO was

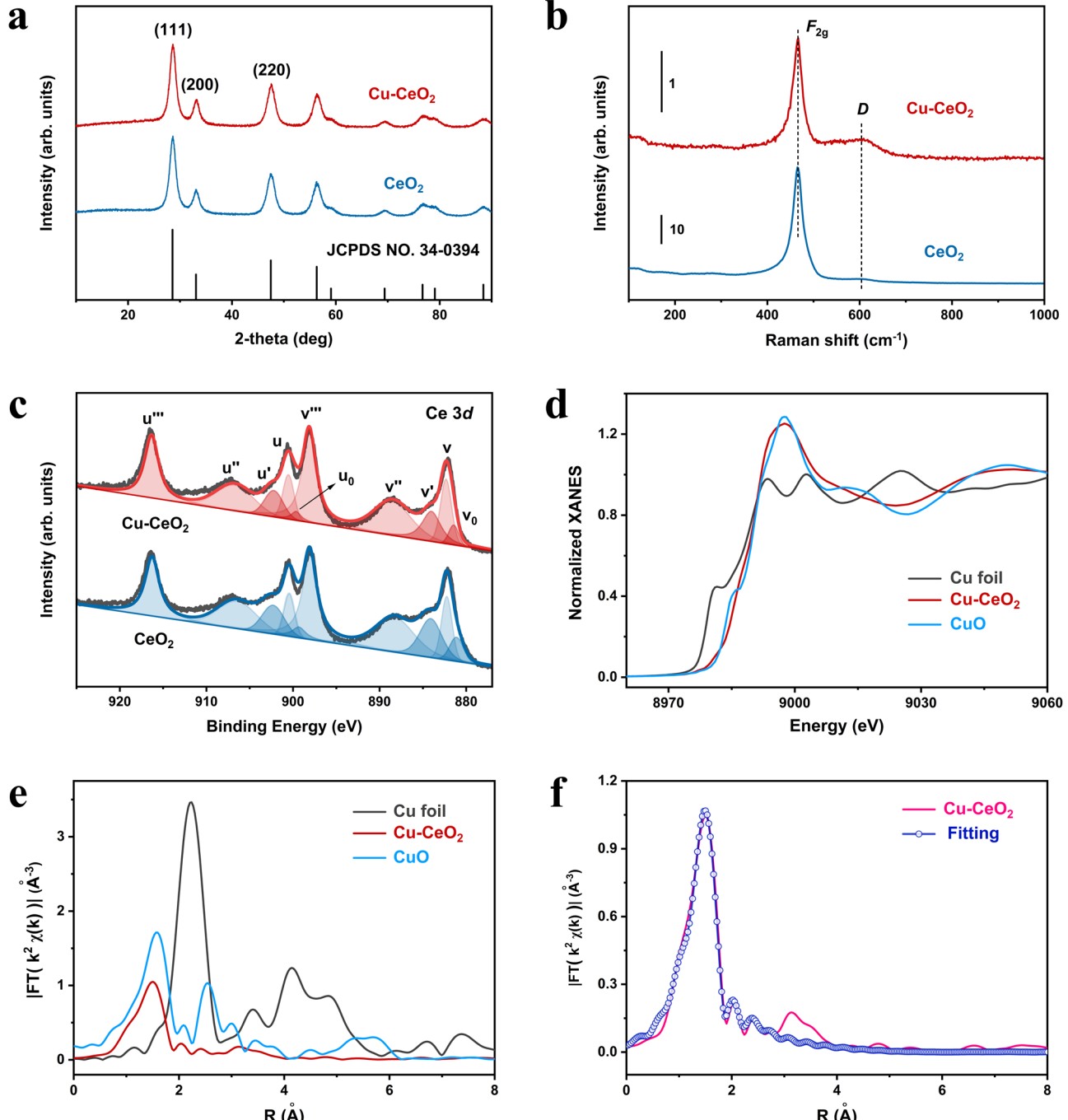

**Fig. 2 | Structural characterizations of Cu-CeO₂ catalyst and reference materials.** XRD patterns (**a**), Raman spectra (**b**) and Ce 3*d* photoelectron profiles (**c**) of the CeO₂ and Cu-CeO₂ catalysts, **d** XANES spectra of Cu foil, CuO and Cu-CeO₂ sample,

**e** Fourier transforms of the Cu K-edge EXAFS oscillations of the materials mentioned above, **f** FT-EXAFS spectra of Cu-CeO₂ sample at the Cu K-edge and the corresponding Fitting results. Source data are provided as a Source Data file.

adsorbed on the Cu⁺ sites[34,39–42]. In addition, the adsorption intensity of the peak gradually increased with the time, and reached saturation adsorption at 480 s with a maximum adsorption peak at 2111 cm⁻¹. Traditionally, the IR band at 2069 cm⁻¹ is regarded as the CO adsorption on Cu⁰ site[40,41]. This indicates that there are no Cu⁰ species in this Cu-CeO₂ catalyst. For CeO₂, two obvious gaseous peak of CO are shown in Supplementary Fig. 9b, which indicating that the CO absorption on parent CeO₂ is absent. XANES spectra at the Ce M₅,₄-edge were normalized and are shown in Supplementary Fig. 10a. The XANES at Ce M₅,₄-edge of CeO₂ based materials correlates with the Ce 3*d*₃/₂ and 3*d*₅/₂ core level transitions into the 4*f* unoccupied electronic state[43]. The intense peak S and peak P represent the tetravalent Ce (4*f*⁰)

while the weak peak R indicates the contribution of trivalent Ce (4*f*¹) states[44,45]. As it vividly shows, Cu-CeO₂ sample exhibits a reduction of Ce³⁺ (peak R) compared with parent CeO₂ while an enhancement of Ce⁴⁺ (peak S). In other words, the Ce³⁺/(Ce³⁺ + Ce⁴⁺) ratio of Cu-CeO₂ is lower than that of CeO₂. The result is in good agreement with XPS results (Fig. 2c, Supplementary Table 1). Supplementary Fig. 10b shows the O K-edge XANES spectra. Three main peaks are attributed to the O 2*p* states that are hybridized with Ce 4*f*, 5*d*(e₉) and 5*d*(t₂₉) states, respectively. The intensity variation is attributed to the structural disorders induced by Cu doping owing to the formation of Cu-O-Ce coordination network[43]. The quantitative FT-EXAFS fitting is conducted to shed light on the structural configuration of Cu (Fig. 2f) and

the corresponding coordination parameters are shown in Supplementary Table 2. It can be seen that the coordination number of the center Cu atom is 3.2 and the mean bond length of Cu-O is about 1.94 Å. Supplementary Fig. 11 shows the q-fitting (inverse Fourier transform) curve of Cu-CeO$_2$, which is consistent with the above R-space fitting results.

## Evaluation of Cu-CeO$_2$ SSE for POD-like and HORAC activity

Multiple enzyme-mimicking activities of Cu-CeO$_2$ were investigated in vitro. Firstly, the POD-like activity of Cu-CeO$_2$ was tested based on the principle that H$_2$O$_2$ could be catalytically decomposed by Cu-CeO$_2$ to generate ·OH, and TMB could be oxidized by ·OH to oxidized-TMB (ox-TMB)[46]. The absorbance of the reaction product at the wavelength of 652 nm was read by a microplate reader. Figure 3a shows that the POD-like activity of Cu-CeO$_2$ was significantly higher than that of pristine CeO$_2$. Furthermore, steady-state kinetic assay was conducted. According to Lineweaver-Burk equation, the Michaelis-Menten constant ($K_m$) of CeO$_2$ and Cu-CeO$_2$ were 24.34 mM and 30.76 mM, respectively. The maximal reaction velocity ($V_{max}$) of CeO$_2$ and Cu-CeO$_2$ were 28.05 nM/s and 166.7 nM/s, respectively, and Cu-CeO$_2$ showed significantly enhanced POD-like activity (Supplementary Fig. 12, Supplementary Table 3). We further investigated the regulation of Cu content on the POD-like catalytic performance of Cu-CeO$_2$ nanozyme. As shown in Supplementary Fig. 12 and Supplementary Table 3, firstly, all Cu-CeO$_2$ SSE exhibited significantly enhanced POD-like activity compared to CuO. In addition, Cu content and the $V_{max}$ of Cu-CeO$_2$ was positively correlated, which may be caused by the variation of Cu sites involved in the catalytic reaction. However, after calculating the turnover rate, we found that as Cu content increased, the turnover rate showed a gradually decreasing trend. This may be due to the presence of more CuO particles in samples with high Cu content compared to those with low Cu content, and CuO may cause a decrease in the dispersion of active centers. It is worth noting that Cu-CeO$_2$ with a theoretical Cu content of 5%, which is also the main sample of this study, can achieve a $V_{max}$ comparable to 10% Cu sample and a turnover rate comparable to 2% Cu sample at the same mass concentration, demonstrating satisfactory catalytic performance. The $V_{max}$ and the turnover rate of 5% Cu-CeO$_2$ was 11.78- and 212.51- higer than CuO nanozyme, respectively, exhibiting significant enhancement of POD-like activity.

Furthermore, the cyclic stability of Cu-CeO$_2$ was tested, the results showed that after 30 catalytic cycles, the POD-like activity of Cu-CeO$_2$ was still comparable with the original nanozyme (Supplementary Fig. 13). The ICP-MS analysis showed that Cu accounts for 4.43 wt % of the Cu-CeO$_2$ sample. Hence, when the concentration of Cu-CeO$_2$ reaches 200 μg/mL, the total content of Cu is 8.858 μg/mL. The content of Cu in the supernatant after cyclic reaction was below 0.500 μg/mL, much lower than the total amount of Cu in the reaction system. In summary, Cu-CeO$_2$ nanozyme has good stability, within our test conditions.

As a reaction substrate, H$_2$O$_2$ can also be catalytically decomposed into O$_2$ by Cu-CeO$_2$, which increases the concentration of dissolved O$_2$ in the liquid environment. The CAT-like activity of Cu-CeO$_2$ was investigated subsequently. Figure 3b shows that the CAT-like activity of Cu-CeO$_2$ was significantly higher than that of pristine CeO$_2$. In addition, O$_2$ can be catalyzed by OXD-like activity of Cu-CeO$_2$ to generate O$_2^-$, which oxidizes TMB to ox-TMB[47]. The absorbance of the reaction product at the wavelength of 652 nm was read by a microplate reader, and Cu-CeO$_2$ showed slightly enhanced OXD-like activity (Fig. 3c). The CAT reaction provides O$_2$, which further promotes the OXD reaction. The SOD-like activity of Cu-CeO$_2$ was also tested. Xanthine-xanthine oxidase system was applied to generate O$_2^-$, which can reduce nitro blue tetrazolium (NBT) to formazan. In the presence of SOD-mimics, O$_2^-$ can be catalytically converted to O$_2$ and H$_2$O$_2$, which further provides H$_2$O$_2$ for POD reaction. As Supplementary

Fig. 14 shows, the SOD-like activity of Cu-CeO$_2$ was also significantly higher than that of pristine CeO$_2$. We further tested the consumption of H$_2$O$_2$ in CeO$_2$ + H$_2$O$_2$ and Cu-CeO$_2$ + H$_2$O$_2$ systems quantitatively. As shown in Supplementary Fig. 15, as the reaction time prolongs, both systems showed a trend of H$_2$O$_2$ consumption, and Cu-CeO$_2$ maintains a higher consumption than CeO$_2$ during long-term reaction process. This may be caused by the enhanced multiple catalytic activities of Cu-CeO$_2$, which accelerate the multiple pathway conversion related to H$_2$O$_2$.

Next, the HORAC activity of Cu-CeO$_2$ was tested. The ·OH generated from H$_2$O$_2$ and Fenton reagent can be transformed into H$_2$O and O$_2$ through HORAC activity, so as to quench the free radical fluorescent probe[48]. As Fig. 3d shows, the fluorescence intensity of CeO$_2$ and Cu-CeO$_2$ groups decreased compared with the reaction baseline, indicating that both nanozymes exerted HORAC activity. However, within 10 min of reaction, the fluorescence intensity of CeO$_2$ group drastically decreased by ~94.31%, compared with baseline, while Cu-CeO$_2$ group only decreased by ~1.92%, and the relative fluorescence intensity was higher than that of the H$_2$O group throughout the entire reaction process, indicating that Cu single-site led to a significant inhibition of HORAC activity of CeO$_2$, and that Cu-CeO$_2$ could accelerate the Fenton-like catalytic process. We further used DMPO as the spin trapping agent and measured the changes of ·OH species quantitatively through EPR. As shown in Supplementary Fig. 16 and Supplementary Table 4, after 10 minutes of reaction, both FeCl$_2$ and FeCl$_2$ + Cu-CeO$_2$ group exhibited typical DMPO-OH signal. The signal intensity of FeCl$_2$ + Cu-CeO$_2$ group was higher than that of FeCl$_2$ group, while the signal of FeCl$_2$ + CeO$_2$ group was almost invisible, which is consistent with the pattern of the fluorescence results. Quantitative calculation showed that the ·OH scavenging rate of CeO$_2$ was 97.26% at 10 min, while the spin concentration of DMPO-OH in FeCl$_2$ + Cu-CeO$_2$ group reached 158.62% of that in FeCl$_2$ group. In addition, we investigated the catalytic effect of the physical mixed system of free Cu$^{2+}$ ions/CuO nanozyme with CeO$_2$ nanozyme. As shown in Supplementary Fig. 17, the initial rate of POD-like reaction of the two physical mixed systems was similar to that of CeO$_2$. As the reaction continued, the substrate conversion extent of the two groups at the end point of the reaction was similar to that of Cu-CeO$_2$. However, DCFH fluorescence detection (Supplementary Fig. 18) showed that for physical mixed systems, only Cu$^{2+}$ + CeO$_2$ group showed significantly higher total ROS generation than CeO$_2$ group, but was still much lower than that of Cu-CeO$_2$ group, indicating that in the non-interacting Cu$^{2+}$/CuO + CeO$_2$ physically mixed system, Cu could not inhibit the HORAC activity of CeO$_2$ effectively. Through the above results, we reaffirm the enhanced POD-like activity of Cu-CeO$_2$ nanozyme, and a prominent inhibitory effect of Cu single sites on the HORAC activity of CeO$_2$. Meanwhile, the interaction between Cu single sites and CeO$_2$ support also plays a key role in the regulation of catalytic activities. Therefore, by the aid of Cu single sites and its interaction with CeO$_2$ support, the effective regulation of the redox catalytic pathways of CeO$_2$ nanozyme was achieved.

Previous studies have shown that structural factors such as Ce$^{3+}$/Ce$^{4+}$ ratio[49], oxygen vacancy[15], defect[11,50], as well as environmental factors such as pH value[51] and ·OH concentration[52], will affect the type of catalytic reaction and catalytic activity of CeO$_2$. Among these factors, Ce$^{3+}$/Ce$^{4+}$ ratio is one of the most important structural factors which determines the POD-like and HORAC activities of CeO$_2$[11,15,49]. Specifically, increasing the proportion of Ce$^{3+}$ can not only improve the POD-like activity of CeO$_2$, but also enhance its HORAC activity. In this study, XPS results indicated that the Ce$^{3+}$/Ce$^{4+}$ ratio of Cu-CeO$_2$ was lower than that of CeO$_2$. Combined with the catalytic activity results, we speculated that the introduction of Cu site inhibits the HORAC activity of CeO$_2$ carrier, and may also lead to the decrease of its POD-like activity. Under the synergistic effect of the intrinsic activity of Cu site, the overall POD-like activity of Cu-CeO$_2$ SSE was maintained.

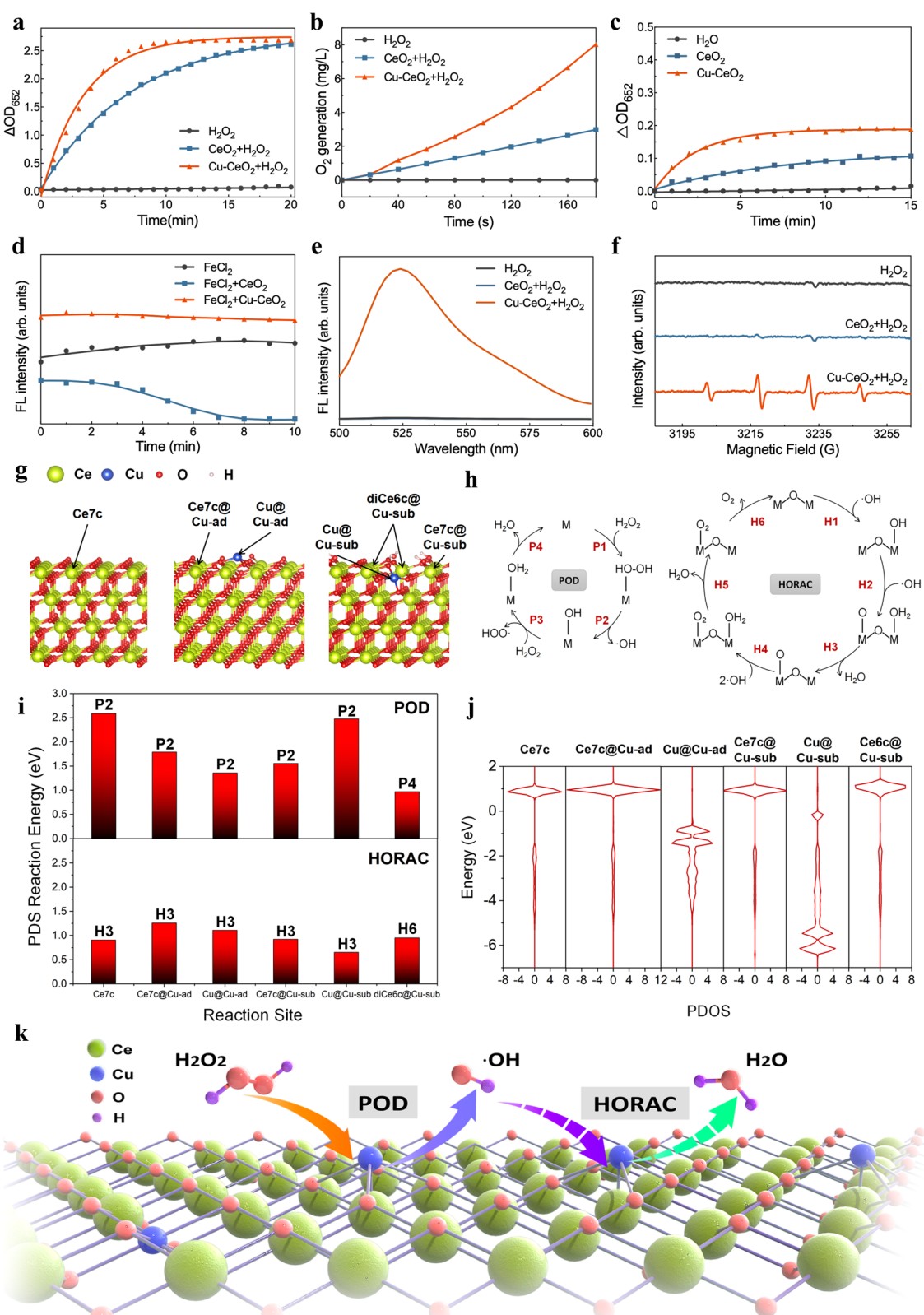

In order to further explore the generation of reactive oxygen species (ROS) in the catalytic decomposition of $H_2O_2$ by Cu-CeO$_2$, 2,7-Dichlorodihydrofluorescein diacetate (DCFH-DA) was adopted as total-ROS detecting fluorescent probe. As Fig. 3e shows, Cu-CeO$_2$ + $H_2O_2$ group produced strong fluorescence signal, $H_2O_2$ group produced weak fluorescence signal, while CeO$_2$ + $H_2O_2$ group exhibited even lower fluorescence than $H_2O_2$ group. These results indicated

that Cu-CeO$_2$ + $H_2O_2$ system can produce abundant ROS efficiently, and CeO$_2$ + $H_2O_2$ system demonstrated an outcome of ROS removal. Terephthalic acid (TA) was adopted as ·OH detecting fluorescent probe. The characteristic fluorescent sprectrum of hydroxyterephthalic acid (TAOH) in Cu-CeO$_2$ group indicated the presence of ·OH (Supplementary Fig. 17). Next, 5,5-dimethyl-1-pyrroline N-oxide (DMPO) was adopted as the spin trap and EPR was used to detect ·OH

**Fig. 3 | Catalytic activities and mechanism of CeO₂ and Cu-CeO₂ nanozymes.** **a** Time-dependent optical density change at 652 nm of 3,3', 5,5'-tetramethylbenzidine (TMB) in POD reactions. **b** Time dependent oxygen generation in CAT reactions. **c** Time dependent optical density change at 652 nm of TMB in OXD reactions. **d** Time dependent fluorescent intensity of 2,7-Dichlorodihydrofluorescein diacetate (DCFH) in HORAC reactions. **e** Fluorescent spectra of DCFH after 10 min reaction with $H_2O_2$ and different nanozymes. **f** Electron Paramagnetic Resonance (EPR) spectra of DMPO-OH after 10 min reaction with $H_2O_2$ and different nanozymes. **g** The calculated model of pristine $CeO_2$ (left), $CeO_2$ with Cu3c added on the 3O atoms on the surface (Cu-ad, middle) and $CeO_2$ with a Ce atom substituted by Cu (Cu-sub, right). The 6 possible reaction sites are highlighted with a tagged arrow. **h** The reaction mechanism of POD and HORAC process, M indicates the metal sites. **i** The calculated PDS reaction energy of POD and HORAC processes for different reaction sites with the exact PDS labeled above the bar. **j** The calculated PDOS, i.e., the d and f band summation of different reaction centers. **k** The proposed mechanism of regulation of catalytic activities by Cu-CeO₂ single-site nanozyme. Source data are provided as a Source Data file.

generation in $Cu\text{-}CeO_2 + H_2O_2$ system. As Fig. 3f shows, no obvious DMPO-OH signal was detected in $H_2O_2$ or $CeO_2 + H_2O_2$ group, while $Cu\text{-}CeO_2 + H_2O_2$ exhibited significant DMPO-OH signal. The catalytic generation of $O_2^{·-}$ by $Cu\text{-}CeO_2$ in methanol was also detected. As Supplementary Fig. 18 shows, among all three groups, only $Cu\text{-}CeO_2$ group showed significant DPMO-$O_2^·$ signal. Combined with the catalytic activity assays above, the results indicated that both $CeO_2$ and $Cu\text{-}CeO_2$ can catalyze the decomposition of $H_2O_2$ to generate ·OH through POD-like activity, and $Cu\text{-}CeO_2$ exerts higher POD-like activity, producing more ·OH. $CeO_2$ may remove ·OH through its high HORAC activity. However, in $Cu\text{-}CeO_2 + H_2O_2$ system, there remain abundant ·OH radicals due to the inhibited HORAC activity of $Cu\text{-}CeO_2$.

## Theoretical analysis

Density functional theory (DFT) is employed to elucidate the underlying mechanism for the boost of ROS generation of $Cu\text{-}CeO_2$. Pristine $CeO_2$ (111) surface (left Fig. 3g) and $Cu\text{-}CeO_2$ (111) surface are built for comparison. Firstly, two configurations of $Cu\text{-}CeO_2$, including one that has a Cu atom directly adhering on the three O atoms on the (111) surface (Cu-ad, middle Fig. 3g), and one that has a Cu atom substitute the surface Ce atom (Cu-sub, right Fig. 3g), are optimized. It should be noted that due to the weak O-binding energy for Cu atom, originally presented surface O3c transformed to O2c in the Cu-sub model, which is with stronger alkalinity and will be protonated in the solution[53]. Thus, the protonated Cu-sub model is finally adopted. Different reaction sites in these systems are considered. For the pristine (111) surface, only one type of reaction site of Ce7c presents. While for the Cu-ad and Cu-sub surface, two and three types of reaction sites are considered, respectively, as shown in Fig. 3g. In the Cu-ad system, both Cu (Cu@Cu-ad) and Ce (Ce7c@Cu-ad) adjacent to Cu are considered. While for the Cu-sub system, the Ce7c site adjacent to the protonated O atom (Ce7c@Cu-sub) and the Cu site (Cu@Cu-sub), along with the double Ce6c site (diCe6c@Cu-sub) that undergoes lattice oxygen mechanism which involves the protonated O2c during the reaction, are considered. DFT optimizations were then performed for the reaction intermediates on different sites. The optimized geometries are illustrated in Supplementary Figs. 21–26.

In this work, a large amount of ·OH radicals are detected in the experiment and are considered as the major disinfection factor. It is crucial to study the activity of POD that generates ·OH, including reaction P1 to P4 in Fig. 3h.

The reaction energy for the potential determining steps (PDS) of different reaction sites are illustrated in the top of Fig. 3i, with the exact PDS labeled on the bars. The detailed reaction energy along the POD pathway are listed in Supplementary Table 5. It can be found that the Ce7c site on the pristine $CeO_2$ (111) surface has a poor POD activity due to the huge energy gap of 2.591 eV during reaction P2 as the PDS. Meanwhile, in the Cu-ad system, the Cu@Cu-ad center served as a catalytic site with a drastically decreased energy of 1.358 eV, while the adjacent Ce7c@Cu-ad sites also have an increased activity with PDS energy as low as 1.792 eV. The influence of Cu in the Cu-sub system are, on one hand, greatly increased the activity of the adjacent Ce7c@Cu-sub and especially the diCe6c@Cu-sub sites, with the PDS energy of 1.556 eV and 0.970 eV, respectively. On the other hand, little change of the activity for Cu@Cu-sub center itself has been calculated (2.479 eV).

These results indicated that Cu single-site can largely promote the overall POD activities with the activity of itself and via activating the adjacent Ce sites.

The electronic structure analysis, namely the projected density of state (PDOS) analysis, is further conducted for different reaction sites to elucidate the exact mechanism for activity promotion. The summation of d and f bands for the reaction sites are shown in Fig. 3j. Firstly, Cu@Cu-ad can serve as the catalytic center in Cu-ad system due to the fact the site is under-coordinated (Cu3c) and that the d band of the center is concentrated near the fermi-level compared with the Ce7c site, which served a stronger interaction between the centers and the oxidized species, stabilized the *OH intermediates and decreased the energy for the PDS (P2). Cu@Cu-sub on the other hand, due to being fully coordinated (Cu4c), the d-band is far away from the fermi-level, weak binding with oxidized species is anticipated and no better performance is calculated. Secondly, Cu single-site also influenced the performance of adjacent Ce. Comparing with the Ce7c plot, a shift to a higher energy can be spotted in Ce7c@Cu-ad and diCe6c@Cu-sub. Such shift is mainly originated from the less coordinated environment of Ce and will contribute to a higher energy of the anti-bond band of the bond between Ce and adsorbed O, further stabilizing the adsorbed *OH structure. In this case, we see a drastic decrement in reaction energy for reaction P2 of $Cu\text{-}CeO_2$. Meanwhile, tighter Ce-O binding facilitates a harder dehydration process and turns reaction P4 into the PDS for the diCe6c@Cu-sub center. For the Ce7c@Cu-sub site, slightly difference could be told from the PDOS from Ce7c, which indicates that on one hand, the introduction of Cu has little influence on the electronic structure for the adjacent Ce7c@Cu-sub sites. On the other hand, the sharp decline for reaction energy in PDS is mainly contributed by the *OH structure that stabilized with the protonated O atom.

It should be noticed that as reported in the literature, $CeO_2$ itself can act as an ROS elimination catalyst to prevent oxidative damage, HORAC mechanism (reaction H1 to H6 in Fig. 3h), as an antagonistic pathway of POD, should also be considered. As depicted in Fig. 3i, the PDS for most reaction sites are the dehydration reaction H3, the reaction energies of which are 0.909 eV for pristine Ce7c. Thus, it is anticipated that as the introduction of Cu to the system, the dehydration process should be hindered due to stronger oxidized species adsorption of the reaction sites on and around Cu single-site (except for Cu@Cu-sub), as discussed in the PDOS analysis. To be specific, the diCe6c@Cu-sub site has an increased PDS energy of 0.954 eV for reaction H6, which inhibited the HORAC process. Likewise, the Cu@Cu-ad, Ce7c@Cu-ad, Ce7c@Cu-sub sites all have worse HORAC performance with PDS energy increased to 1.110, 0.924 and 1.258 eV. The PDS energy on Cu@Cu-sub site is 0.652 eV, which indicated a better HORAC activity. Yet, considering the weak interaction between adsorbed ·OH and Cu@Cu-sub center, the HORAC reaction tends to take place on the Ce sites instead of Cu, which limits the promotion effect for Cu@Cu-sub center promotion to the HORAC processes.

In all, Cu single-site influences the reaction activity on and around itself. Except for the drastic energy decrement of POD, the inhibition of HORAC processes also made contributions to the overall ROS generation performance (Fig. 3k).

To elucidate the possible influence of small CuO cluster on the surface, additional DFT calculations were performed. According to previous literatures, small Cu nano-clusters on $CeO_2$ facets prefer to be in the form of monolayers, a $(CuO)_3$-$CeO_2$ model was thus established[54,55]. The stable planar $(CuO)_3$ cluster with similar symmetry of $CeO_2$ (111) surface was loaded. Considering the aqueous environment of the actual experimental condition, water molecule was added and found dissociated spontaneously on the top of the CuO cluster under the strong influence of 3 under coordinated Cu, forming the hydrated site as shown in Supplementary Figs. 27, 28. Due to the structural complexity of the cluster, there are two possible reaction sites, i.e., the $Cu_3$ site and the $Cu_2Ce$ site which surround the *OH intermediate. Both HORAC and POD-like mechanisms were calculated on these sites. The results indicated that for POD-like pathway, the energy consumption of the PDS are sharply reduced to 0.796 eV ($Cu_3$) and 0.915 eV ($Cu_2Ce$) mainly due to the strong interaction of the intermediates (*OH) with multiple coordination atoms. Thus, $(CuO)_3$ cluster boosted the activity of ·OH radical generation. While for the HORAC pathway, the strong binding of $O_2$ with multiple atoms further increase the energy consumption of the PDS ($O_2$ desorption) to 1.081 eV ($Cu_3$) and 1.015 eV ($Cu_2Ce$), which deactivated the HORAC process. The above results suggest that small Cu nano-clusters also possess oxidant activities. Based on the steady-state kinetic results of POD-like activities of Cu-$CeO_2$ samples with different Cu contents in Supplementary Table 3, when the Cu content increased from 2 to 5%, the $V_{max}$ increased by ~2.13 folds, and the turnover rate showed a slight decrease, indicating that potential existence of small amount of Cu nanoclusters in Cu-$CeO_2$ sample may contribute to the overall POD-like activity of the nanozyme. However, when the Cu content continued to increase to 10%, the $V_{max}$ did not increase correspondingly, but remained comparable to that of the 5% sample. Meanwhile, the turnover rate drastically decreased by ~2.10 folds, suggesting that the presence of a large number of Cu nano-clusters may actually inhibit the overall catalytic activity of the nanozyme. The phenomenon we observed is consistent with that reported in previous literature[56].

## In vitro antibacterial performance of Cu-$CeO_2$ SSE

After confirming the catalytic activities and underlying ROS generation mechanism of Cu-$CeO_2$, in vitro and in vivo experiments were carried out to investigate its antibacterial properties. Methicillin-resistant MRSA and *E. coli* were chosen as representative strains of Gram-positive and Gram-negative bacteria, respectively. The bacterial solution was treated in six groups: PBS, $CeO_2$, Cu-$CeO_2$, $H_2O_2$, $CeO_2 + H_2O_2$ and Cu-$CeO_2 + H_2O_2$. After different treatments, the antibacterial effect of each group was evaluated by plate colony-counting. The results showed that except for the Cu-$CeO_2 + H_2O_2$ group, there was no significant difference colony numbers between the other groups and the control group ($P > 0.05$) (Fig. 4a, b, g, i). Nanozyme solely showed no obvious antibacterial effect. For MRSA, there was no significant difference in the reduction of colony number after $H_2O_2$ or $CeO_2 + H_2O_2$ treatment (~0.12-log), while Cu-$CeO_2 + H_2O_2$ group completely sterilized ~6.85-log MRSA and showed superior antibacterial properties. Similarly, for *E. coli*, after $H_2O_2$ treatment, the colony number decreased by ~0.91-log. It's noteworthy that after $CeO_2 + H_2O_2$ treatment, the colony number decreased by ~0.56-log, which was even worse than that of $H_2O_2$ group. However, Cu-$CeO_2 + H_2O_2$ group completely sterilized ~7.70-log *E. coli*, reversing the inhibited antibacterial effect of $CeO_2$. The bacterial solution of each group was stained and observed using fluorescence microscope, and the results are shown in Fig. 4c, d and Supplementary Fig. 27. Almost all bacteria in groups without $H_2O_2$ were alive, with only a few dead ones. There were a few dead bacteria in $H_2O_2$ group and $CeO_2 + H_2O_2$ group, most of which were living bacteria. For Cu-$CeO_2 + H_2O_2$ group, no obvious living bacteria were observed. The antibacterial effect of each group was further confirmed by SEM. Except that Cu-$CeO_2 + H_2O_2$ group

showed obvious loss of bacterial membrane integrity, the bacterial morphology of the other groups was intact or slightly wrinkled (Fig. 4e, f, Supplementary Fig. 28). $H_2O_2$ has been widely used for debridement and disinfection of various infected wounds clinically due to its broad-spectrum antibacterial properties[57]. However, high concentrations of $H_2O_2$ can be harmful to normal human tissues. In addition, the antibacterial ability of $H_2O_2$ is relatively weak, and different bacteria perform varied sensitivity to $H_2O_2$[58]. By the aid of the intrinsic peroxidase-like activity of Cu-$CeO_2$, the application of Cu-$CeO_2 + H_2O_2$ system can achieve better bactericidal performance using much lower concentration of $H_2O_2$ than that in clinic (2.5% ~ 3.5%).

To further confirm ROS generation in the Cu-$CeO_2 + H_2O_2$ antibacterial system, DCFH staining was performed. As shown in Supplementary Figs. 28, 29, for both MRSA and *E. coli*, the fluorescence of DCFH-DA in Cu-$CeO_2 + H_2O_2$ group are more obvious than $CeO_2 + H_2O_2$ group. In addition, the fluorescence was well co-localized with nanozyme-bacteria composite, while planktonic bacteria showed little fluorescence, indicating that the onset of ROS-mediated sterilization in Cu-$CeO_2 + H_2O_2$ antibacterial system is likely to be located on the nanozyme-bacteria interface.

We further explored the rules of bactericidal changes of $CeO_2$ and Cu-$CeO_2$ nanozymes at different concentrations. For MRSA, the arbitrary combinations of $CeO_2 + H_2O_2$ had no significant antibacterial effect, while Cu-$CeO_2 + H_2O_2$ groups showed obviously positive correlation between the bactericidal effect and the concentration of nanozyme and $H_2O_2$ (Fig. 4h, j, Supplementary Figs. 30, 31). For *E. coli*, due to its high sensitivity to $H_2O_2$, the colony number showed a decreasing tendency with the increase of $H_2O_2$ concentration. Notably the antibacterial effect gradually deteriorated with the increase of $CeO_2$ concentration, which was ever worse than that of $H_2O_2$ with the same concentration. Cu-$CeO_2 + H_2O_2$ group still showed a positive correlation between antibacterial effect and nanozyme and $H_2O_2$ concentration. The phenomenon observed in our study is similar to that reported by Zhu et al.[14]. According to their report, the synthesized spherical $CeO_2$ nanozyme with a particle size of ~150 nm exhibited optimistic POD-like activity at pH = 4.0–6.0. However, the bactericidal effect of $CeO_2 + H_2O_2$ on *E. coli* in PBS (pH = 4.0) was significantly weaker than that of using $H_2O_2$ alone, and no significant ·OH was detected in the $CeO_2 + H_2O_2$ system. Therefore, they speculated that the ROS scavenging ability of $CeO_2$ hinders the decomposition of $H_2O_2$ to generate ·OH, and its POD-like activity did not contribute to the antibacterial effect of the $CeO_2 + H_2O_2$ system. Combined with the former characterizations of catalytic activities and ROS generation in nanozyme + $H_2O_2$ system, it can be inferred that the low POD-like activity and high HORAC activity of $CeO_2 + H_2O_2$ system lead to a relatively low amount of ·OH available for sterilization in the environment. Besides, the concentration of $H_2O_2$ substrate decreased as POD reaction occurred, resulting in the weakening of its antibacterial effect. In other words, $CeO_2$ protected bacteria from $H_2O_2$ and ·OH. In contrast to $CeO_2$, Cu-$CeO_2 + H_2O_2$ system showed high POD activity and low HORAC activity. The amount of ·OH in the environment was enough to meet the needs of sterilization.

Human gingival fibroblasts (hGFs) and human periodontal ligament stem cells (hPDLScs) were adopted to test the cytotoxicity of the nanozymes on normal human cells. The CCK-8 results showed that there was no significant difference in the relative activity of cells in each group with the increase of nanozyme concentration ($P > 0.05$, Supplementary Fig. 32a, b). Meanwhile, Live/Dead cell staining showed that there was no significant number of red stained dead cells in each group (Supplementary Fig. 32c). The maximum concentration of nanozymes used in cell safety test was 200 μg/mL, which was much bigger than the dose required for the antibacterial experiments, indicating that Cu-$CeO_2$ nanozyme had good biosafety.

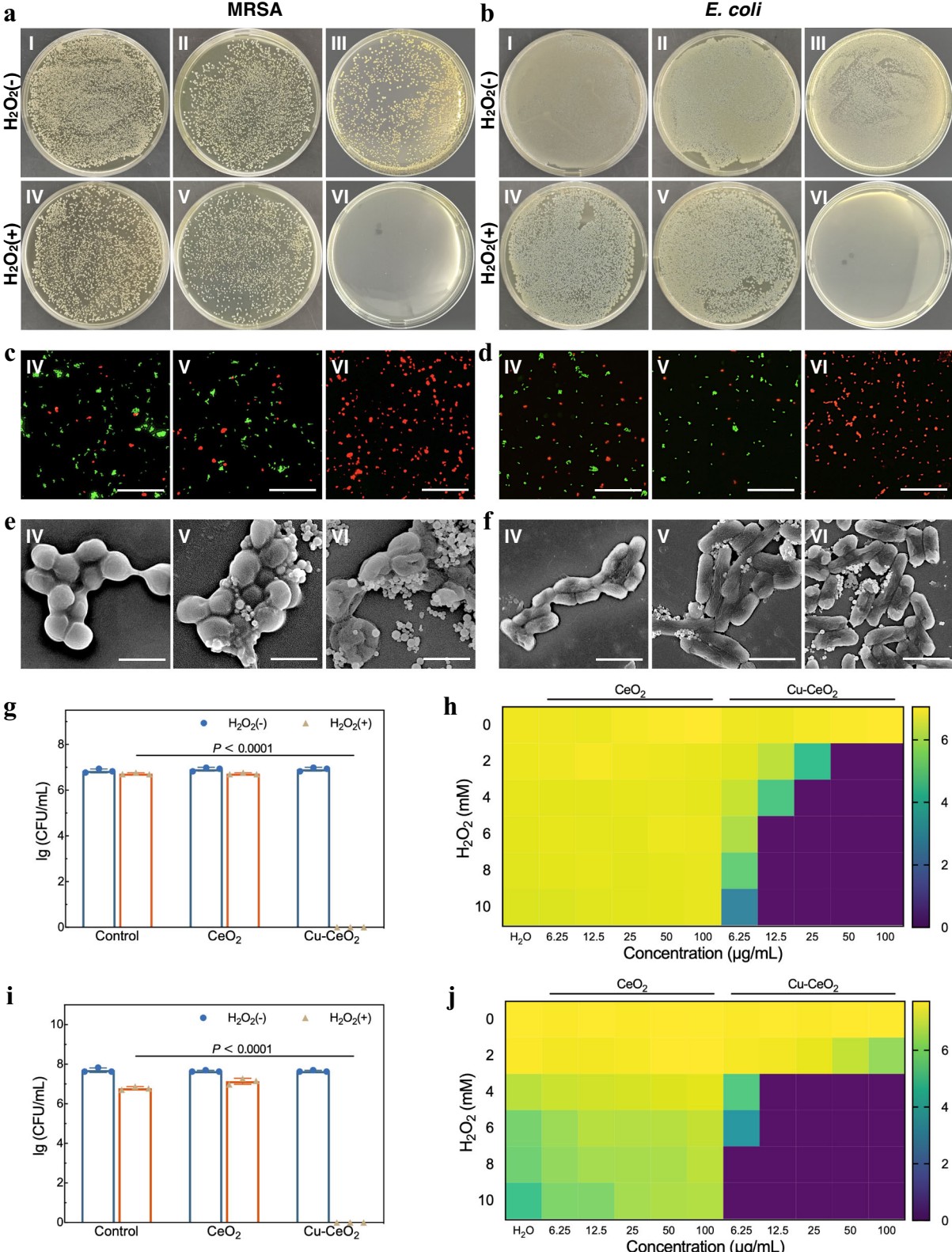

**Fig. 4 | In vitro antibacterial properties of CeO$_2$ and Cu-CeO$_2$ nanozymes.**
Bacterial colonies, fluorescent images (scale bar: 50 μm) and SEM images (scale bar: 1 μm) of MRSA (**a**, **c**, **e**) and *E. coli* (**b**, **d**, **f**) after grouped treatment (I: Phosphate buffered saline (PBS), II: CeO$_2$, III: Cu-CeO$_2$, IV: H$_2$O$_2$, V: CeO$_2$ + H$_2$O$_2$, VI: Cu-CeO$_2$ + H$_2$O$_2$). Logarithm of colony forming unit (CFU) countings of MRSA (**g**) and *E. coli* (**i**) after grouped treatment. Logarithm of CFU countings of MRSA (**h**) and *E. coli* (**j**) after treatment with different concentrations of nanozymes and H$_2$O$_2$. A representative image of three replicates from each group is shown. Data are presented as mean values +/− standard deviation, $n = 3$ biologically independent replicates. Significance was calculated by two-sided Student's *t*-test. Source data are provided as a Source Data file.

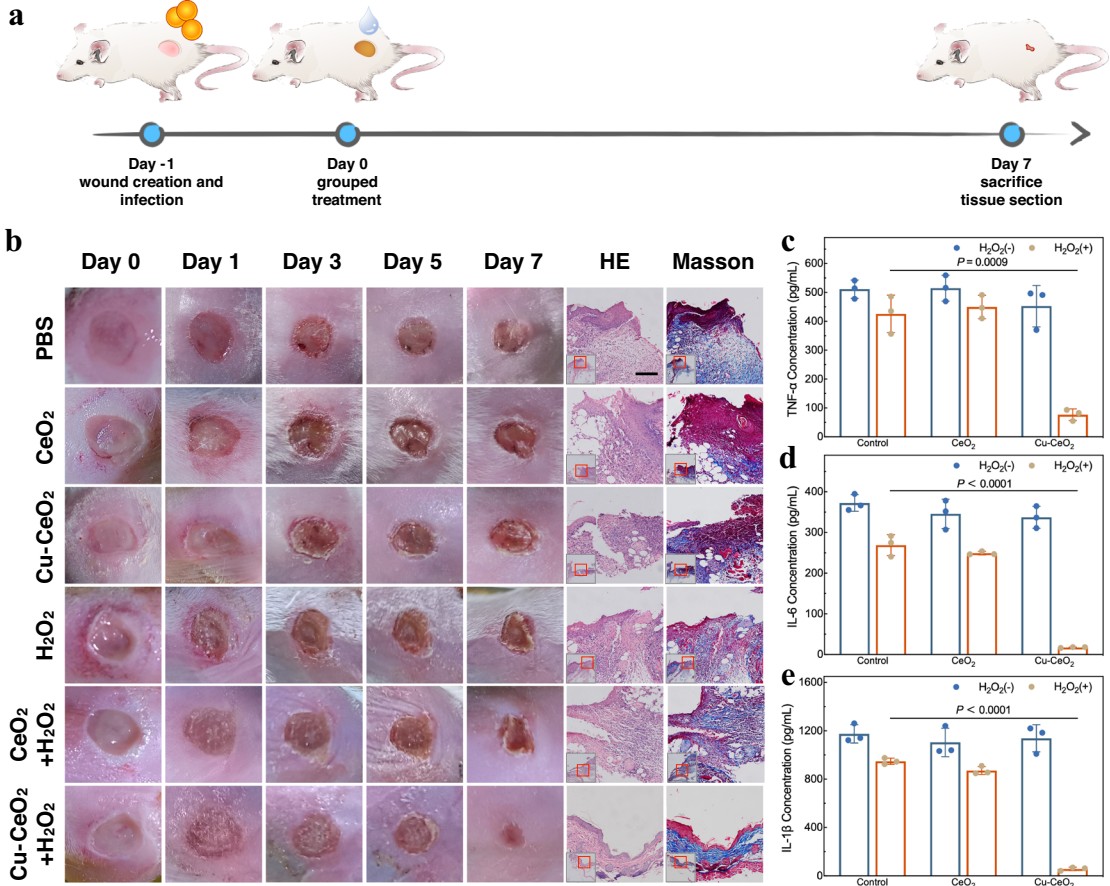

**Fig. 5 | In vivo antibacterial properties of CeO₂ and Cu-CeO₂ nanozymes.**
**a** Schematic procedure of skin wound infection model and treatment.
**b** Photographs, HE staining and Masson staining of the infected skin wounds after grouped treatment. Concentrations of TNF-α (**c**), IL-6 (**d**) and IL-1β (**e**) in skin tissue homogenates after grouped treatment. A representative image of three replicates from each group is shown. Data are presented as mean values +/− standard deviation, n = 3 biologically independent replicates. Significance was calculated by two-sided Student's *t*-test. Scale bar: 500 μm. Source data are provided as a Source Data file.

## In vivo safety of Cu-CeO₂ SSE

To test the in vivo safety of the nanozymes, PBS, CeO₂, and Cu-CeO₂ was intravenously injected into balb/c mice. The mice were observed for 3 days consecutively. As Supplementary Fig. 33 shows, there was no significant differences among the average body weight of the three groups. In addition, the blood routine test results at day 1 and day 3 after administration showed no statistical difference between each group in terms of the average white blood cell (WBC) and red blood cell (RBC) counts (*P* > 0.05, Supplementary Fig. 34). The hematoxylin-eosin (HE) staining results of the main organs on day 1 and day 3 indicate no significant histological differences between CeO₂, Cu-CeO₂ and PBS group (Supplementary Fig. 35). In summary, both CeO₂ and Cu-CeO₂ nanozymes showed good in vivo safety.

## In vivo antibacterial performance of Cu-CeO₂ SSE

According to previous literature, the pH value of normal intact skin tissue is weakly acidic (pH = 4−6), while in the case of wound infection, due to inflammatory stimulation, the local pH value tends to be weakly alkaline (pH≈7.4)[59]. It was confirmed through in vitro experiments that Cu-CeO₂ catalytically decomposed H₂O₂ efficiently to generate ROS under pH = 7.4, achieving excellent antibacterial effects and good cyclic stability. These inspired us to further explore the therapeutic effect of Cu-CeO₂ + H₂O₂ on in vivo infected skin wounds. For in vivo bactericidal experiment, we established a mouse skin wound infection model, and the schematic procedure is shown in Fig. 5a. A full-thickness skin defect with a diameter of ~6 mm was made on the back of BALB/c mice, and the wound was contaminated with MRSA. After 24 h of infection, the wounds were treated in six groups (PBS, CeO₂, Cu-CeO₂, H₂O₂, CeO₂ + H₂O₂ and Cu-CeO₂ + H₂O₂, *n* = 4), and the healing of skin wounds was observed daily. As Fig. 5b shows, 24 h after MRSA inoculation (Day 0), pyogenic infection occurred locally in the wounds of each group. After grouped treatments, the wounds in each group showed a trend of scab formation and contraction, and Cu-CeO₂ + H₂O₂ group showed the fastest wound healing speed (Supplementary Fig. 36). On day 7, the wounds in Cu-CeO₂ + H₂O₂ group were completely closed and the scabs fell off. The mice were sacrificed on day 7, and tissue sections of the skin around the wound were prepared. The results of HE staining showed that the epithelial structure of Cu-CeO₂ + H₂O₂ group was approximately intact and continuous with no signs of infection, while the epithelium of the other groups was discontinuous. The epithelial defects showed varying degrees of deep dyeing inflammatory cells infiltration and extensive unstructured necrosis. Masson's trichrome staining was used to observe the distribution of collagen fibers in skin tissues of each group. There were a large number of coarse, blue stained newly formed collagen fibers under the intact epithelium in Cu-CeO₂ + H₂O₂ group, and the collagen fibers were uniformly distributed. In the other groups, however, the collagen fibers were scattered, inconsistent in thickness and disordered in structure. A certain amount of skin tissue around the wound of each group were taken to prepare homogenates, and the expression of three important pro-inflammatory factors, tumor necrosis factor α (TNF-α), interleukin-6 (IL-6) and interleukin 1β (IL-1β) was detected by

enzyme-linked immunosorbent assay (ELISA) kit to reflect the severity of infection[60]. After $H_2O_2$ or $CeO_2 + H_2O_2$ treatment, the level of inflammatory factors in skin tissue homogenates decreased compared with the control group, but the difference was not statistically significant ($P > 0.05$). However, the level of inflammatory factors in Cu-$CeO_2 + H_2O_2$ group was significantly lower than the control group ($P < 0.001$) (Fig. 5c–e). Immunohistochemistry staining was further performed to evaluate the expression of TNF-α and IL-1β in the skin tissues, and the results are shown in Supplementary Figs. 37–40, which were consistent with the ELISA results. The above results indicated that Cu-$CeO_2 + H_2O_2$ system showed good antibacterial activity in vivo and effectively reduced the inflammatory response caused by wound infection.

## Discussion

In this study, a Cu-$CeO_2$ SSE was synthesized by a facile method. The enzyme-mimicking activities of $CeO_2$ were regulated by Cu single site modification. Foreign Cu sites not only reduced the reaction energy of the PDS drastically during POD-like process at Ce site, but also increased its reaction energy of the PDS during HORAC process, resulting in higher POD-like activity and lower HORAC activity of Cu-$CeO_2$ than pristine $CeO_2$. Therefore, Cu-$CeO_2 + H_2O_2$ system successfully reversed the protective effect of $CeO_2$ on bacteria and demonstrated superior antibacterial properties. This study not only expands the existing antibacterial SSE family, but also reveals the regulation mechanism of metal single site on carriers with complex catalytic activities, which has important enlightenment for designing and developing smart nanocatalysts in the future.

On the other hand, $CeO_2$, as an excellent antioxidant nanozyme, has received widespread attention in promoting injury healing[61,62]. In this study, although the ·OH scavenging capacity of Cu-$CeO_2$ was selectively inhibited, the scavenging acitivites of common ROS (i.e., $H_2O_2$ and $O_2^{·-}$) in the injury microenvironment were enhanced compared to $CeO_2$. This may be one of the advantages of Cu-$CeO_2$ in treating infected wounds. Specifically, since Cu-$CeO_2$ generates a large amount of ·OH only when applied simultaneously with $H_2O_2$, and does not generate excess toxic ROS in the absence of $H_2O_2$, when $H_2O_2$ is completely consumed, the residual Cu-$CeO_2$ nanozyme may continue to scavenge endogenous ROS in the wound microenvironment. It is clear that within the scope of this study, Cu-$CeO_2 + H_2O_2$ system plays an important role in the early anti-infectious process. However, its ability to accelerate wound healing in the subsequent antioxidant and tissue healing processes has not been thoroughly studied, which is also a limitation of this study.

## Methods

### Chemicals and materials

All chemicals were used as received without further purification. Cerium nitrate hexahydrate (Ce(NO$_3$)$_3$·6H$_2$O, 99.95%), copper nitrate trihydrate (Cu(NO$_3$)$_2$·3H$_2$O, 99.99%), TMB (C$_{16}$H$_{20}$N$_2$, 98%), Ferrous chloride (FeCl$_2$, 99.5%), Sodium acetate (CH$_3$COONa, NaAc, 99.0%) and TA (C$_8$H$_6$O$_4$, 99.0%) were obtained from the Aladdin chemical reagent company. Sodium carbonate (Na$_2$CO$_3$, 99.8%), ethylene glycol ((CH$_2$OH)$_2$, 99.5%) and acetic acid (CH$_3$COOH, 99.5%) were purchased from Sinopharm Chemical Reagent Co., Ltd. Deionized (DI) water from Milli-Q System (resistivity: 18.2 MΩ·cm, Millipore, Billerica, MA) was used in all experimental processes. CCK-8 and DMPO were obtained from Dojindo Laboratories Co., Ltd. LIVE/DEAD™ Viability/Cytotoxicity Kit, LIVE/DEAD™BacLight™ Bacterial Viability/Counting Kit and IL-1 beta Polyclonal Antibody (P420B) were obtained from ThermoFisher Scientific Co., Ltd. Mouse TNF-α, IL-6, IL-1β ELISA kit were obtained from 4ABio Co., Ltd. Recombinant anti-TNF alpha antibody (ab307164) was obtained from Abcam Co., Ltd.

### Synthesis of Cu-$CeO_2$

In a typical preparation of $CeO_2$ nanosphere, 1.0 g of Ce(NO$_3$)$_3$·6H$_2$O was dispersed in a mixed solvent of water (1 mL), acetic acid (1 mL) and ethylene glycol (25 mL) to form a uniform solution. Then, the mixture was transferred into a steel reactor followed by a solvothermal reaction at 180 °C for 200 min. Then the precipitate was washed by centrifuging at 19,319 × g for 10 min with water for several times. After drying, the resultant products were calcined in air at 400 °C for 4 h with a heating rate of 2 °C min⁻¹ to acquire the final $CeO_2$ nanospheres. 200 mg of the obtained $CeO_2$ powder was dispersed in 30 mL of deionized water and sonicated for 30 min to form a uniform dispersion. After that, copper nitrate aqueous solution was slowly added to the above suspension under stirring conditions. The copper content in weight percent was 5%. Five minutes later, 0.5 M sodium carbonate aqueous solution was added dropwise to ensure the pH value of the solution at ca. 9. Then the precipitates were aged at ambient temperature for 6 h, followed by centrifugation (16,099 × g, 3 min) and washed with water and ethanol several times. Subsequently, the product was dried at 80 °C overnight to acquire Cu-$CeO_2$ precursor. The precursor was calcined at 400 °C for 4 h in air to get the sample Cu-$CeO_2$ air. The Cu-$CeO_2$ air sample was then annealed at 250 °C for 1 h under a 5% $H_2$/95% Ar atmosphere to obtain the target product Cu-$CeO_2$.

### Characterization

Transmission electron microscopy (TEM, Hitachi H-7650) was used to obtain the morphologies of the samples. The high-resolution TEM (JEOL JEM-2100F) and elemental mappings were operated at an accelerating voltage of 200 kV. Atomically resolved HAADF-STEM images were recorded on a JEM-ARM200F transmission electron microscope with a spherical aberration corrector operated at 200 kV. Field-emission scanning electron microscopy measurements were carried out on a SU-8200 scanning electron microscope. The powder XRD patterns were obtained on a Japan Rigaku RU-200b X-ray diffractometer using Cu-Kα radiation with λ = 1.5406 Å. The theoretical X-ray diffraction pattern was simulated using MAUD version 2.8 (Materials Analysis Using Diffraction). XPS experiments were conducted on a ULVAC PHI Quantera microscope. Raman measurements were carried out with a Renishaw inVia micro-Raman spectrometer with a 532 nm excitation laser. All of the DRIFTS spectra were collected by using a Bruker Vertex 70 FTIR spectrometer. The XAFS of Cu K-edge spectra were measured on the beamline BL11B station in Shanghai Synchrotron Radiation Facility (SSRF). The acquired XAFS data was processed using the ATHENA module in the Demeter 0.9.25 software package. Soft XAS spectra of Ce $M_{4,5}$-edge and O K-edge were measured at soft X-ray magnetic circular dichroism end station (XMCD) of National Synchrotron Radiation Laboratory (NSRL) in University of Science and Technology of China (USTC).

### Measurements

This research complies with all relevant ethical regulations, and the study protocols are approved by the biomedical ethics committee of Peking University.

### Peroxidase-like activity of Cu-$CeO_2$

The POD-like activity was tested according to the principle that $H_2O_2$ was catalytically decomposed by Cu-$CeO_2$ and TMB was oxidized in 0.1 M NaAc/HAc buffer (pH = 4.5). TMB was dissolved in dimethyl sulfoxide (DMSO) at the concentration of 41.6 mM before use. The generation of water-soluable ox-TMB was measured by a multimode plate reader (PerkinElmer EnSpire™), and the absorbance at 652 nm was recorded. For time-dependent catalytic assay, the final concentrations of Cu-$CeO_2$, $H_2O_2$ and TMB in the system were 50 μg/mL, 44.1 mM and 1040 μM, respectively. The result was recorded every

60 s, 1200 s in total. For steady-state kinetic assays, the system contains 50 μg/mL Cu-CeO$_2$, 208 μM TMB with 15-150 mM H$_2$O$_2$. $K_m$ and $V_{max}$ were calculated by Lineweaver-Burk equation. The turnover rate of the nanozymes were calculated as follow:

$$\text{Turnover rate} = V_{max}/[E] \qquad (1)$$

Where $[E]$ represents the total concentration of Cu atoms in the reaction system.

### Catalase-like activity of Cu-CeO$_2$

The CAT-like activity was tested according to the principle that H$_2$O$_2$ was catalytically decomposed by Cu-CeO$_2$ to generate O$_2$ in PBS buffer (pH = 7.4). The final concentrations of Cu-CeO$_2$ and H$_2$O$_2$ in the system were 20 μg/mL and 0.6% (w/v). The concentration of dissolved O$_2$ was measured with a dissolved oxygen meter (JPSJ-606L, Leici China).

### Oxidase-like activity of Cu-CeO$_2$

The OXD-like activity of Cu-CeO$_2$ was tested according to the principle that O$_2$ was catalytically converted by Cu-CeO$_2$ and TMB was oxidized in 0.1 M NaAc/HAc buffer (pH = 4.5). TMB was dissolved in DMSO at the concentration of 41.6 mM before use. The generation of water-soluable ox-TMB was measured by a multimode plate reader, and the absorbance at 652 nm was recorded. The final concentrations of Cu-CeO$_2$ and TMB were 50 μg/mL and 1040 μM. The result was recorded every 60 s, 900 s in total.

DMPO was used as the spin trap of O$_2^{\cdot-}$, and the DMPO-O$_2^{\cdot-}$ spin in methanol was measured by EPR (Bruker Magnettech ESR5000) at the following parameters: microwave power of 2.5 mW, modulation field frequency of 100 kHz, amplitude of 0.2 mT, central magnetic field of 316.5 mT and scanning speed of 20 mT/min. A typical experimental system includes 10 mM DMPO and 50 μg/mL Cu-CeO$_2$. The reaction system was incubated at room temperature for 10 min.

### Superoxide-dismutase-like activity of Cu-CeO$_2$

According to the manufacturer's instructions, the SOD activity detection kit was used to detect the SOD-like activity of Cu-CeO$_2$. Xanthine/xanthine oxidase system was used to generate superoxide anion (O$_2^{-}$), which reduced nitro blue tetrazolium (NBT) to formazan. O$_2^{\cdot-}$ could be removed through the SOD-like activity of Cu-CeO$_2$ so that the formation of methyl was inhibited. The absorbance at 560 nm was recorded.

### H$_2$O$_2$ consumption of Cu-CeO$_2$

H$_2$O$_2$ consumption of the nanozymes were determined by spectrophotometry. In a classic assay, the final concentrations of nanozyme and H$_2$O$_2$ in the system were 100 μg/mL and 10 mM, respectively. After incubated at 37 °C for different time, samples were taken and the concentration of H$_2$O$_2$ was measured using a hydrogen peroxide content assay kit (Solarbio BC3595).

### Hydroxyl radical scavenging ability of Cu-CeO$_2$

FeCl$_2$ was used as the Fenton reagent to decompose H$_2$O$_2$ to generate ·OH, and DCFH-DA was chosen as the free radical fluorescence probe. Through the HORAC activity of CeO$_2$ and Cu-CeO$_2$, ·OH can be removed so as to quench the fluorescence. The final concentrations of nanozymes, DCFH-DA, FeCl$_2$ and H$_2$O$_2$ were 100 μg/mL, 5 μM, 4 mM and 4 mM, respectively. Immediately after the reaction, the fluorescence intensity was measured with a multimode plate reader ($\lambda_{em}$ = 490 nm, $\lambda_{ex}$ = 525 nm) and recorded every minute, and the final fluorescence intensity was compared with the initial fluorescence intensity (baseline) of each group. The relative change of fluorescence intensity was calculated as follow:

$$\text{Relative change} = |F_0 - F_t|/F_0 \times 100\% \qquad (2)$$

Where $F_0$ represents the initial fluorescence intensity (baseline), and $F_t$ represents the fluorescence intensity after certain time of reaction.

Quantitative analysis of ·OH scavenging was performed using EPR (Bruker Magnettech ESR5000). DMPO was used as the spin trap of ·OH, and FeCl$_2$ was used as the Fenton reagent to decompose H$_2$O$_2$. The final concentrations of nanozymes, DMPO, FeCl$_2$ and H$_2$O$_2$ were 50 μg/mL, 10 mM, 1 mM and 1 mM, respectively. The detection parameters are as follow: microwave power, 2.5 mW; modulation field frequency, 100 kHz; amplitude, 0.2 mT; central magnetic field, 336.5 mT and scanning speed, 20 mT/min.

### Detection of ROS

DCFH-DA was used as the fluorescence probe of ROS. ROS generation during the process of catalytic decomposition of H$_2$O$_2$ in PBS was measured and recorded by a multimode plate reader ($\lambda_{em}$ = 490 nm, $\lambda_{ex}$ = 525 nm). A typical experimental system included 5 μM DCFH-DA, 50 μg/mL Cu-CeO$_2$ and 10 mM H$_2$O$_2$. The reaction system was incubated at room temperature in the dark for 10 min.

DMPO was used as the spin trap of ·OH, and the catalytic decomposition of H$_2$O$_2$ by Cu-CeO$_2$ in 0.1 M NaAc / HAc buffer (pH = 4.5) was measured and recorded by EPR (JEOL JES FA200) at the following parameters: microwave power of 1 mW at the frequency of 9.0–9.1 GHz, modulation field frequency of 100 kHz, amplitude of 0.1 mT, time constant of 0.1 s, central magnetic field of 322.5 mT and scanning speed of 10 mT/min. A typical experimental system includes 100 mM DMPO, 50 μg/mL Cu-CeO$_2$ and 10 mM H$_2$O$_2$. The reaction system was incubated at room temperature for 10 min.

### Cyclic stability of Cu-CeO$_2$

200 μg/mL Cu-CeO$_2$ and 10 mM H$_2$O$_2$ were mixed in PBS. After certain time of reaction, the precipitate was recovered by centrifugation at $20,627 \times g$. Cu-CeO$_2$ after 10, 20 and 30 cycles were collected and POD-like activities of the samples were tested. The content of Cu element in the supernatant after reaction was determined using ICP-MS (Thermo Scientific XSeries 2).

### Calculation details

The DFT calculations were conducted with the Vienna Ab-initio Simulation Package (VASP 6.1.0)[63,64] with the Projected Augmented Waves (PAW) basis and Perdew-Burke-Ernzerhof (PBE)[65] exchange-correlation functional. DFT + U correction with the effective U-J value of 5 eV[66] and 3 eV[67] were adopted to describe the exchange interaction of Ce 4$f$ and Cu 3$d$ orbitals, respectively. Spin-polarization was applied throughout the calculations along with the Grimme's DFT-D3[68] method accounting for the long-range van der Waals interaction. A 4-layered $3 \times 3$ CeO$_2$ 111 supercell was built to represent the pristine CeO$_2$ surface. 20 Å vacuum space was added along the z direction to avoid interaction between the adjacent layers. Two Cu-doped CeO$_2$ surface, namely Cu-ad and Cu-sub were built on the basis of the pristine surface to investigate the exact doping configuration. The 400 eV energy cutoff and $3 \times 3 \times 1$ Gamma centered k-point mesh was adopted during the calculation. The geometries for the reaction intermediates were optimized with the bottom two CeO$_2$ layers fixed until the residual force less than 0.03 eV/Å.

### In vitro antibacterial properties of Cu-CeO$_2$

E. coli and MRSA were cultured in brain heart infusion (BHI) broth to logarithmic growth phase, and the absorbance of bacterial solution at 630 nm wavelength was measured by a microplate reader. The bacterial solution was centrifuged at $9167 \times g$ for 3 min, and the precipitation was resuspended with PBS buffer. In a typical experimental system, the concentrations of bacteria, nanozyme and H$_2$O$_2$ were ~10$^6$ CFU/mL, 25 μg/mL and 4 mM for MRSA, and ~10$^7$ CFU/mL, 25 μg/mL and 4 mM for E. coli, respectively. After incubation at 37 °C for 1 h,

gradient dilutions were inoculated on BHI agar dish, incubated in an air incubator at 37 °C overnight, and the colonies were counted.

Bacterial live/dead staining (ThermoFisher LIVE/DEAD BacLight Bacterial Viability Kit) were performed according to the manufacturer's instructions. SYTO-9 and propidium iodide (PI) were added to the reaction system after co-culture, incubated in dark for 15 min. The bacterial solution was observed with a fluorescence microscope ($\lambda_{em} = 480$ nm, $\lambda_{ex} = 500$ nm for SYTO-9, and $\lambda_{em} = 535$ nm, $\lambda_{ex} = 615$ nm for PI).

For SEM observation, the bacterial solution after co-culture was fixed with 4% paraformaldehyde-PBS and dehydrated with gradient ethanol. The samples underwent gold sputtering at 10 mA and 40 mBar vacuum, and the morphology of the bacteria was observed.

### In vitro cytotoxicity of Cu-CeO$_2$

HGFs and hPDLScs were inoculated in 96 well plates at $10^4$/well and cultured with Dulbecco's modified Eagle medium (DMEM) containing 10% fetal bovine serum (FBS). The nanozymes were added for co-culture, and the final concentrations were 0, 3, 12, 50, 100 and 200 μg/mL, respectively. After incubation for 24 h, proliferation medium containing 10% Cell Counting Kit-8 (CCK-8) reagent was added. After incubation for 1 h, the absorbance at 450 nm was measured using a microplate reader. For live/dead cell staining, after co-cultured with nanozymes, the cells were washed with PBS, and calcein AM/PI (Keygen Biotech KGAF001) were added according to the manufacturer's instructions. After incubation for 15 min, the cells were observed with a fluorescence microscope (OLYMPUS U-RFL-T $\lambda_{em} = 490$ nm, $\lambda_{ex} = 515$ nm for calcein AM, and $\lambda_{em} = 535$ nm, $\lambda_{ex} = 615$ nm for PI).

### In vivo experiments

The animal experiments were carried out under the approval of the biomedical ethics committee of Peking University (approval number: LA2021473). Mice were kept in constant temperature (22 °C), constant humidity (55%) and cyclic lighting (12 h light/12 h dark). For in vivo safety analysis, eighteen 6-week-old BALB/c female mice were purchased from SiPeiFu Biotechnology Co., Ltd, Beijing, and were injected with PBS, CeO$_2$, Cu-CeO$_2$ through i.v., respectively. The body weights were recorded consecutively for 3 days. At day 1 and day 3 after administration, peripheral blood and main organs were collected for further analysis. For in vivo antibacterial experiment, twenty-four 6-week-old BALB/c female mice were purchased from SiPeiFu Biotechnology Co., Ltd, Beijing. MRSA was cultured to logarithmic growth stage in BHI liquid, and the bacterial solution was adjusted to $10^6 \sim 10^7$ CFU/mL with PBS. After anesthesia, a full-thickness wound with a diameter of ~6 mm was made on the dorsal skin of the mice through aseptic operation. MRSA solution was injected locally to make an infected wound model. After 24 h of infection, the wounds were treated in groups (PBS, CeO$_2$, Cu-CeO$_2$, H$_2$O$_2$, CeO$_2$ + H$_2$O$_2$ and Cu-CeO$_2$ + H$_2$O$_2$), and covered with Band-Aids. The Band-Aids were changed every 24 h. The concentrations of nanozymes and H$_2$O$_2$ were 25 μg/mL and 4 mM, respectively. The wound healing was observed, and the wound size were measured. On the 7th day after infection, the mice were sacrificed and local skin tissue was obtained. 50 mg samples were obtained from each tissue and lysed with Radio immunoprecipitation assay buffer (Ripa): Phenylmethanesulfonyl fluoride (PMSF) = 100:1 mixed lysate to prepare tissue homogenate. The supernatant was obtained under $14,000 \times g$ centrifugation for 10 min at 4 °C. The protein concentration was detected by Bicinchoninic acid (BCA) method. IL-6, IL-1, and TNF-α contents were detected using ELISA kit. The skin tissues of each group were fixed with 10% neutral formalin solution for 24 h. The tissues were dehydrated and paraffin embedded. Tissue sections were made and HE Masson and immunohistochemistry staining were used for histological analysis. For immunohistochemistry staining, recombinant anti-TNF alpha antibody (dilution of 1:1000) and IL-1 beta Polyclonal Antibody (dilution of 1:500) were used as primary antibodies. Detailed information (supplier name, catalog number) about the antibodies are listed in Reporting Summary.

### Statistical analysis

The experimental data were analyzed using IBM SPSS Statistics 24.0 software. One-way ANOVA was used for comparisons, and Games-Howell test was used for post hoc multiple comparisons.

### Reporting summary

Further information on research design is available in the Nature Portfolio Reporting Summary linked to this article.

## Data availability

All data are available within the Article and Supplementary Files (Supplementary Information, Supplementary Data 1), or available from the corresponding authors upon request. Source data are provided with this paper.

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

## Acknowledgements

This study was supported by the National Natural Science Foundation of China (51972003, Y.W., 52271127, Y.W., 22325101, D.W.). We thank the BL14W1 station in Shanghai Synchrotron Radiation Facility (SSRF) and 1W1B station in Beijing Synchrotron Radiation Facility (BSRF) for XAFS measurement.

## Author contributions

P.J., L.Z. and X.L. contributed equally to this research. P.J., L.Z., Y.W. and D.W. conceived the idea and designed the project. P.J. and L.Z. designed the experiments. L.Z. carried out the biological experiments and ana-lyzed the data. X.L. contributed to theoretical calculation in this paper and wrote the part of calculations. C.Y., P.Z., T.T., Y.W. and D.W. co-write the manuscript. All authors discussed the results and commented on the manuscript.

## Competing interests
The authors declare no competing interests.
