## [Peer Review File · Nature Communications]

Reviewers' Comments:

Reviewer #1:

Remarks to the Author:

The ability of nanozymes to exhibit multiple enzyme-like activities offers the potential for cascade catalysis. However, the existence of competitive catalytic reactions often leads to suboptimal anti-oxidation or pro-oxidation performance of nanozymes. Therefore, it is crucial to rationally regulate the activities of nanozymes. In this study, the authors successfully synthesized a Cu-CeO₂ nanozyme where the activities of CeO₂ were modulated by incorporating Cu single-atoms. This modulation resulted in an increase in peroxidase-like (POD) activity and a decrease in the free radical scavenging capacity of Cu-CeO₂. Consequently, the Cu-CeO₂ nanozyme exhibited significant antibacterial properties in the presence of H₂O₂. The topic of this study is intriguing. I recommend that the article be published after the authors addressing the following concerns.

1. Why did the author not mention the enhancement of POD activity in the title, while emphasizing its significance in the article?
2. Why did the author choose to anchor copper single-atom sites, why not other metals?
3. The direct data of the Cu single-atom in Cu-CeO₂ nanozyme is insufficient.
4. The content of Cu in Cu-CeO₂ nanozyme should better be provided.
5. The content "CeO₂, with a fluorite-like cubic structure which each metal ion is close-packed by eight O²⁻ ions" is incorrect.
6. CeO₂ nanozyme often show multienzyme-like activities, such as SOD, POD, CAT, and OXD-like activities. I strongly recommend the authors to discuss the fundamental factors which determine the kinds of enzyme-like activity. This is very important to design nanozyme structures that enhance or inhibit a given enzyme-like activity.
7. The formation process of Cu single-atom in Cu-CeO₂ nanozyme should be discussed.
8. Single-atom nanozymes have exhibited excellent catalytic performance. The catalytic activities of Cu-CeO₂ should be quantitative analyzed and compared with other previous reported nanozymes.
9. line 164 and 192, 562 nm should be 652 nm.
10. Figure S3, the measurement of 0.28nm does not match the description in the text of 0.27nm.
11. In Figure 2b, how to prove that this is an oxygen defect and not other defects?
12. Please supplement the result analysis for Figure S5.
13. "Also, Ce M4,5 absorption edge soft XAS spectra (Figure S9a) and O K-edge XANES spectra (Figure S9b) for CeO₂ and Cu-CeO₂ show no obvious difference" Please provide further explanation.
14. The Cu CeO₂ air catalyst refers to the product obtained after heating the composite in air? Please explain or draw the synthesis steps and the obtained product in Figure 1a
15. Figure S11b, in the steady-state kinetic assay, the substrate concentration establishment is not reasonable. Usually, a suitable substrate concentration range should straddle the K_m value and range roughly from 0.5 × K_m to 5 × K_m.
16. In Figure S12 (Relative cyclic stability of CeO₂ and Cu-CeO₂), there is only a group of data.

Reviewer #2:

Remarks to the Author:

The manuscript by Jiang et al. presents a systematic study on improved peroxidase-like activity and hydroxyl radical antioxidant capacity of Cu-CeO₂ nanozymes with respect to CeO₂ counterpart. The material characterization is quite extensive, but raises a few concerns as listed below. The DFT-based mechanistic study provides important insights into observed activity enhancements. Yet simulations consider only single-atom Cu species as active sites, while larger Cu species (clusters and nanoparticles) can also be present in the synthesized system. Possible contribution of those metal sites is not discussed. Overall I recommend this work for publication after resolving the questions below:

- 1) The title reads "Ceric Dioxide". As there is no "ceric" element in periodic table, the title must be updated with "cerium dioxide" or "ceria".
- 2) STEM EDX imaging with such a large pixel size (in the range of few nm) as shown in Fig S4 cannot provide sufficient evidence for atomically dispersed nature of copper species. Effectively these data only suggests that Cu is homogeneously distributed over the sample on the scale of few

nm, but not necessarily on the atomic level. The EXAFS data indeed suggests that Cu species are highly-dispersed, but does not rule out the presence of Cu clusters and/or nanoparticles of sub-nm to few-nm size. Recent work (10.1063/5.0008748) demonstrates that amorphous CuO nanoparticles below 3 nm can exhibit an EXAFS spectrum virtually resembling atomically dispersed CuO species. Thus EXAFS and STEM EDX (without atomic-resolution EELS/EDX elemental mapping) are not sufficient to unambiguously conclude the single-atom nature of metallic sites. Authors should consider use of CO-probe molecule FTIR studies to confirm single-atom nature of metal sites. Otherwise, the authors should refrain from claiming that all Cu species are atomically dispersed or represented by single sites.

3) The authors performed XPS study of Ce state in both CeO₂ and Cu-CeO₂ samples. I would like to complement the authors on successful efforts in fitting complex Ce 3d spectra. For completeness, I would recommend to refer to the literature which describes the used fitting model. As the authors suggest on the basis of Raman data that addition of Cu influences the concentration of defects/oxygen vacancies (and thus Ce³⁺ concentration), it is highly recommended to report the actual Ce³⁺/Ce⁴⁺ ratios derived from the Ce3d fitting shown in Fig 2c. From the visual assessment I can see that the intensity of Ce³⁺ peaks in CeO₂ is higher than in Cu-CeO₂, which would contradict the claim on concentration of defect based on Raman data. Could the authors provide the actual numbers and discuss these data quantitatively?

4) The much higher ROS generation activity of Cu-CeO₂ with respect to pure CeO₂ shown in Fig 3e is striking. As for some test the reaction media is acidic, is it possible that Cu leaches away into solution and acts as a homogeneous catalyst? Had ICP or any other analytical method been used to conclude on possible Cu dissolution? Could the authors perform a control measurement with pure CuO under identical experimental conditions (Fig 4c)? Please also fix a typo in Fig 4d – I assume it is H₂O₂ instead of H₂O in EPR data labelling.

5) The DFT modelling and mechanistic study solely rely on single-atom structure of Cu species in prepared nanozymes. As discussed above, from the presented data it is difficult to conclude that all Cu species are atomically dispersed. In reality, clusters and nanoparticles of transition metals often accompany single-atom species in the sample. It would be worthwhile if authors could provide some calculations (see e.g., 10.1039/D1SC01201K) or at least discuss a possible contribution of small CuO clusters to the POD-like and HORAC activity.

6) The discussion of in vitro experiments is too technical and contains a lot of experimental details (e.g., concentrations) which should be listed in the methods section. It would improve readability if authors would focus on the main findings of these tests.

Reviewer #3:

Remarks to the Author:

This study developed a single-atom copper modified CeO₂ nanozyme with multiple catalytic activities and inhibited free radical scavenging capacity for antibacterial application. The catalytic mechanism regulated by the interaction between single-atom Cu and CeO₂ carrier was explored using experimental and theoretical methods. The antibacterial performance of Cu-CeO₂ systems in vitro and in vivo was also confirmed. However, I do not believe that the novelty and quality of this manuscript meet the high standard of Nature Communications.

Specific comments:

1. CeO₂ nanosphere was prepared in this study. However, the illustrations of CeO₂, Cu-CeO₂ precursor and Cu-CeO₂ SASE in Figure 1a look like a hollow structure, which could readily confuse readers.

2. Multiple catalytic properties of Cu-CeO₂ were investigated in Figure 3. It was concluded that the higher CAT, SOD, as well as OXD-like activities of Cu-CeO₂ may also contribute to reproducing H₂O₂ through cascade reactions. Does it mean that the consumption of H₂O₂ by Cu-CeO₂ is slower than that of CeO₂? The consumption of H₂O₂ by Cu-CeO₂ and CeO₂ should be provided?

3. On Line 203, "the fluorescence intensity of CeO₂ group drastically decreased by ~94.31%, while Cu-CeO₂ group merely decreased by ~1.92% (Figure S14)". Figure S14 shows relative clearance rate of ·OH based on the result in Figure 3d. However, the fluorescence intensity of Cu-CeO₂ group was even higher than that of H₂O group. Please explain it and provide the calculating formula of relative clearance rate of ·OH.

4. Only ·OH generation was confirmed by EPR characterization. What about O₂•-?

5. It is still not clear that what the valence state of Cu is. The statements, "the isolated single Cu atoms bear a high oxidation valence state in Cu-CeO₂ sample" and "Cu atoms are present as a monodispersed state in Cu-CeO₂ sample", seem to be inconsistent.

6. Relative cell viability of hGF and hPDLSc was investigated after treatment with different concentrations of Cu-CeO₂. However, Cu-CeO₂ + H₂O₂ system was used for the in vivo antibacterial and wound healing experiments. Generally, the level of ROS is high in the infected wounds, which delays the wound healing process. Therefore, antioxidant agents have attracted much attention in the field of wound dressing. Please indicate the advantage of Cu-CeO₂ + H₂O₂ system for infected wound healing.

Dear Editors and Reviewers:

Thanks for your letter and for the reviewers' comments concerning our manuscript entitled "Tuning Activity of Ceric Dioxide for Antibacterial Application by Anchoring Copper Single-Atom Site via Inhibiting Free Radical Scavenging Capacity" (No. NCOMMS-23-26628). Those valuable and professional comments are very useful for us to revise and improve our manuscript. We have studied all the comments carefully and additional experiments, analysis and revisions were performed to improve our manuscript. In addition, we have supplemented *in vivo* safety experiments based on the editor's suggestions (Line 27-30, Page 19, Line 1-5, Page 20 in the revised main article, Supplementary Fig. 33-35 in the revised Supplementary Information).

The main corrections in the paper and a point-by-point response to the concerns are as following (The corrections are highlighted with yellow color in the revised main article and Supplementary Information).

For Reviewer #1:

Reviewer #1: The ability of nanozymes to exhibit multiple enzyme-like activities offers the potential for cascade catalysis. However, the existence of competitive catalytic reactions often leads to suboptimal anti-oxidation or pro-oxidation performance of nanozymes. Therefore, it is crucial to rationally regulate the activities of nanozymes. In this study, the authors successfully synthesized a Cu-CeO₂ nanozyme where the activities of CeO₂ were modulated by incorporating Cu single-atoms. This modulation resulted in an increase in peroxidase-like (POD) activity and a decrease in the free radical scavenging capacity of Cu-CeO₂. Consequently, the Cu-CeO₂ nanozyme exhibited significant antibacterial properties in the presence of H₂O₂. The topic of this study is intriguing. I recommend that the article be published after the authors addressing the following concerns.

Thank you for your affirmation of our manuscript. These valuable and professional suggestions are very helpful for us to improve our manuscript. We have already conducted sufficient experiments and made detailed clarifications and modifications based on your comments.

Comments 1: Why did the author not mention the enhancement of POD activity in the title, while emphasizing its significance in the article?

Reply 1: Thanks for your suggestions. The title has been replaced by "Tuning Oxidant and Antioxidant Activities of Ceria by Anchoring Copper Single-Site for Antibacterial Application".

This has been revised in our revised manuscript and supplementary information.

Comments 2: Why did the author choose to anchor copper single-atom sites, why not other metals?

Reply 2: Thank you for your kind reminders. We have supplemented the reason why we chose to anchor copper single sites, as described in the manuscript: “Cu agent, as a long-standing antibacterial agent, has achieved excellent antibacterial effects through electrostatic adsorption, ion permeation, and disruption of bacterial redox homeostasis.^{1,2} In recent years, it has attracted widespread attention in simulating natural oxidase and peroxidase for antibacterial purposes.^{3,4} However, traditional Cu antibacterial agents often possess a high content of Cu, which not only causes waste of Cu catalytic sites in the core, but also poses a risk of causing damage to normal cells after precipitation in the form of Cu ions in practical applications.⁵ The above factors encourage us to seek safer and more efficient Cu antibacterial materials to solve the problems of poor utilization and stability of traditional Cu antibacterial agents.”

Reference:

1. C. P. Guo, F. Cheng, G. L. Liang, *et al. Chem. Eng. J.*, **2022**, 435, 134915.
2. C. H. Dong, W. Feng, W. W. Xu, *et al. Adv. Sci.*, **2020**, *7*, 2001549.
3. Y. Fan, S. G. Liu, Y. Yi, *et al. ACS Nano.*, **2021**, *15*, 2005-2037.
4. X. P. Liu, Z. Q. Yan, Y. Zhang, *et al. ACS Nano.*, **2019**, *13*, 5222-5230.
5. X. Y. Lu, S. S. Gao, H. Lin, *et al. Adv. Mater.*, **2020**, *32*, 2002246.

Line 13-21, Page 3 in the revised main article.

Comments 3: The direct data of the Cu single-atom in Cu-CeO₂ nanozyme is insufficient.

Reply 3: Since Cu has an atomic number ($Z = 29$) lower than Ce ($Z = 58$), it is difficult to distinguish Cu SAs simply in terms of brightness or darkness as shown in the HAADF-STEM image in Figure 1f. We have performed CO-probe molecule FTIR measurements to investigate the nature of Cu metal sites. In situ DRIFTS was used to investigate the adsorption ability of Cu-CeO₂. As shown in Supplementary Fig. 9, Cu-CeO₂ catalyst exhibits a band around 2105 cm⁻¹, assigned to the linear CO adsorbed on Cu⁺ sites (Cu⁺-CO), indicating that CO was adsorbed on the Cu⁺ sites (Ref. 1-5). Additionally, the adsorption intensity of the peak gradually increased with the time, and reached saturation adsorption at 480s with a maximum adsorption peak at 2111cm⁻¹. Traditionally, the IR band at 2069 cm⁻¹ is regarded as the CO adsorption on Cu⁰ site (Ref. 1, 2). This indicates that there are Cu⁺ species in this Cu-CeO₂ catalyst. Besides, XANES profiles in Figure 2d and FT-EXAFS spectra in Figure 2e also indicate that there is no formation of Cu-Cu bond. Based on the steady-state kinetic results of POD-like activities of Cu-CeO₂ samples with different Cu contents in Supplementary Table 3, when the Cu content increased from 2% to 5%, the V_{\max} increased by ~2.13 folds, and the turnover rate showed a slight decrease, indicating the potential existence of small amount of Cu nanoclusters in our Cu-CeO₂ sample.

Supplementary Fig. 9. In situ DRIFTS study of CO adsorption on Cu-CeO₂. (2% CO/Ar flow rate, 30 mL min⁻¹; catalyst mass, 100 mg; temperature, 30 °C)

Reference:

1. L. Q. Kang, B. L. Wang, A. T. Guntner, *et al. Angew. Chem. Int. Edit.* **2021**, 60, 14420-14428.
2. H. X. Liu, S. Q. Li, W. W. Wang, *et al., Nat. Commun.*, **2022**, 13, 867.
3. W. W. Wang, W. Z. Yu, P. P. Du, *et al., ACS Catal.*, **2017**, 7, 1313-1329.
4. W. Z. Yu, W. W. Wang, S. Q. Li, *et al., J. Am. Chem. Soc.*, **2019**, 141, 17548-17557.
5. A. L. Chen, X. J. Yu, Y. Zhou, *et al. Nat. Catal.*, **2019**, 2, 334-341.

This has been revised in manuscript: Line 15-16, Page 7; Line 1-6, Page 8; Line 9-17, Page 16. Supplementary Fig. 9, Supplementary Table 3 in Supplementary Information.

Comments 4: The content of Cu in Cu-CeO₂ nanozyme should better be provided.

Reply 4: The Cu content of Cu-CeO₂ nanozyme is 4.43 wt % as determined by inductively coupled plasma optical emission spectroscopy (ICP-OES).

Line 20-21, Page 9 in the revised main article.

Comments 5: The content “CeO₂, with a fluorite-like cubic structure which each metal ion is close-packed by eight O²⁻ ions” is incorrect.

Reply 5: Thank you for your suggestions. CeO₂ has a fluorite-like cubic structure which close-packed cerium atoms are coordinated with eight O²⁻ ions (Ref. 1,2).

The inappropriate expressions have been revised in our revised manuscript: Line 26-27, Page 2.

Comments 6: CeO₂ nanozyme often show multienzyme-like activities, such as SOD, POD, CAT, and OXD-like activities. I strongly recommend the authors to discuss the fundamental factors which determine the kinds of enzyme-like activity. This is very important to design nanozyme structures that enhance or inhibit a given enzyme-like activity.

Reply 6: Thank you for your rigorous consideration. We have supplemented discussion on various factors that determine the activity and type of the enzyme-like activities of CeO₂. As discussed in the manuscript, “Previous studies have shown that structural factors such as Ce³⁺/Ce⁴⁺ ratio (Ref. 1), oxygen vacancy (Ref. 2), defect (Ref. 3,4), as well as environmental factors such as pH value (Ref. 5) and ·OH concentration (Ref. 6), will affect the type of catalytic reaction and catalytic activity of CeO₂. Among these factors, Ce³⁺/Ce⁴⁺ ratio is one of the most important structural factors which determines the POD-like and HORAC activities of CeO₂ (Ref. 1-3). Specifically, increasing the proportion of Ce³⁺ can not only improve the POD-like activity of CeO₂, but also enhance its HORAC activity. In this study, XPS results indicated that the Ce³⁺/Ce⁴⁺ ratio of Cu-CeO₂ was lower than that of CeO₂. Combined with the catalytic activity results, we speculated that the introduction of Cu site inhibits the HORAC activity of CeO₂ carrier, and may also lead to the decrease of its POD-like activity. Under the synergistic effect of the intrinsic activity of Cu site, the overall POD-like activity of Cu-CeO₂ SSE was maintained.”

Reference:

1. Y. Xue, Q. F. Luan, D. Yang, *et al. J. Phys. Chem. C.*, **2011**, 115, 4433-4438.
2. Y. Y. Ma, Z. M. Tian, W. F. Zhai, *et al. Nano Res.*, **2022**, 15, 10328-10342.
3. M. Chen, X. C. Zhou, C. Xiong, *et al. ACS Appl. Mater. Interfaces.*, **2022**, 14, 21989-21995.
4. V. Seminko, P. Maksimchuk, G. Grygorova, *et al. Chem. Phys. Lett.*, **2021**, 767, 138363.
5. H. Y. Ma, Z. Liu, P. Koshy, *et al. ACS Appl. Mater. Interfaces.*, **2022**, 14, 11937-11949.
6. M. Lu, Y. Zhang, Y. W. Wang, *et al. ACS Appl. Mater. Interfaces.*, **2016**, 8, 23580-23590.

This has been discussed in manuscript: Line 12-21, Page 12.

Comments 7: The formation process of Cu single-atom in Cu-CeO₂ nanozyme should be discussed.

Reply 7: CeO₂ mesoporous spheres with high surface areas, uniform size distributions, and well-defined pore topologies have been developed as an ideal substrate to anchoring Cu. After synthesizing CeO₂, Ce³⁺, oxygen vacancies (O_v) or other defects are created on the surface. Thus, foreign metal ions could be adsorbed and deposited on the surface in the presence of alkaline solutions. Owing to the Cu-O-Ce interaction, Cu species are present as a highly dispersive state in Cu-CeO₂ sample after calcined in air.

This has been discussed in manuscript: Line 13-17, Page 4.

Comments 8: Single-atom nanozymes have exhibited excellent catalytic performance. The catalytic activities of Cu-CeO₂ should be quantitative analyzed and compared with other previous reported nanozymes.

Reply 8: We have supplemented quantitative analysis on the catalytic activities of Cu-CeO₂, and compared with CuO nanozyme. We calculated the turnover rate of Cu-CeO₂ and CuO nanozymes, as shown in Supplementary Table 3, the V_{max} and the turnover rate of 5% Cu-CeO₂ (the main sample of this study) was 11.78- and 212.51- fold higher than CuO nanozyme, respectively, exhibiting significant enhancement of POD-like activity.

Supplementary Table 3 POD-like kinetic parameters of CeO₂ and Cu-CeO₂. (K_m: Michaelis-Menten constant, V_{max}: maximal reaction velocity)

Catalyst	V _{max} (nM/s)	K _m (mM)	E (M)	Turnover rate(/s)
CeO ₂	28.05	24.34	-	-
CuO	14.15	65.86	6.24×10 ⁻⁴	2.27×10 ⁻⁵
2% Cu-CeO ₂	78.34	35.94	1.47×10 ⁻⁵	5.33×10 ⁻³
5% Cu-CeO ₂	166.7	30.76	3.46×10 ⁻⁵	4.82×10 ⁻³
10% Cu-CeO ₂	161.4	75.83	7.03×10 ⁻⁵	2.30×10 ⁻³

This has been revised in manuscript: Line 5-17, Page 9; Line 17-19, Page 23.
Supplementary Table 3 in Supplementary Information.

Comments 9: line 164 and 192, 562 nm should be 652 nm.

Reply 9: I'm sorry we made a mistake. The expression has been revised. We have corrected 562 nm to 652 nm.

Line 27, Page 8; Line 11, Page 11 in the revised main article.

Comments 10: Figure S3, the measurement of 0.28 nm does not match the description in the text of 0.27 nm.

Reply 10: Thanks for your reminders, we made the mistake as you pointed out. The description in the text has been revised.

Comments 11: In Figure 2b, how to prove that this is an oxygen defect and not other defects?

Reply 11: Thank you for your suggestions. Generally speaking, Raman spectra are employed to detect the defect sites in CeO₂ samples. When CeO₂ is synthesized, Ce³⁺ and oxygen vacancies (O_v) are created on the surface. Traditionally, the band centered at ~460 cm⁻¹ is characteristic of the triply degenerate Raman active mode of cubic fluorite structure of ceria (F_{2g}) while the bands at 500 to 600 cm⁻¹ are assigned to a defect-induced mode (D). The intensity of these two spectra is different, and we have made modification in Figure 2b. The softening of the F_{2g} mode is accompanied by the appearance of a broad feature centered around 600 cm⁻¹. (Ref. 1) The Raman mode is extremely sensitive to changes in the oxygen stoichiometry around the Ce atom. For pure CeO₂, the D mode vibration is produced by the oxygen vacancy caused by the presence of Ce³⁺. However, for doped CeO₂, it is misleading to simply refer to the peak at ca. 600 cm⁻¹ as a response signal to O_v. In addition to oxygen vacancies, point defects and the increasing ordering level can also contribute to the variation of D band (Ref. 2, 3). As pointed out by a special study which focused on doped CeO₂, the appearance of D band is due to the formation of defect species in Ce-O coordination, which leads a consequence that the vibration signal of Ce-O cannot be cancelled in all directions. (Ref. 2) The defects can be divided into an α defect at 560 cm⁻¹ and a β defect at 600 cm⁻¹. The former band was assigned to defect species including an oxygen vacancy, while the latter was assigned to the formation of the MO₈-type complex but did not contain any oxygen vacancies. For β defect, the oxygen vacancy concentration is constant while the increasing disordering level of the sample can also lead to D vibrational modes. Thus the weak peak at about 600 cm⁻¹ in the Raman spectrum of our Cu-CeO₂ sample can be attributed to the β-peak signal mentioned in the above literature. Since there exists a Cu-O-Ce coordination structure in Cu-CeO₂ catalyst, the interaction between Cu-O and Ce-O is not equal, which also leads to the difference in the vibration of Ce-O in different directions. It's reported by Huang and his coworkers that the isolated Cu ions tend to preferentially deposit at the defective sites of CeO₂, likely the surface oxygen vacancies (Ref. 4). Indeed, Oh illustrates that atomically dispersed copper atoms would cause the content of surface V_O and Ce³⁺ to decrease, which is also consistent with our XPS and EPR results. (Ref. 5) Overall, the presence of the D mode cannot be used as direct evidence for the increased concentration of oxygen vacancy.

Reference:

1. J. S. Elias, N. Artrith, M. Bugnet, *et al.* *ACS Catal.* **2016**, 6, 1675-1679.
2. L. Li, F. Chen, J. Q. Lu, *et al.* *J. Phys. Chem. A*, **2011**, 115, 7972-7977.
3. M. Sarkar, M. R. Rajkumar, S. Tripathy, *et al.* *Mater. Res. Bull.*, **2012**, 47, 4340-4346.
4. Y. X. Gao, Z. H. Zhang, Z. R. Li, *et al.* *Chin. J. Catal.*, **2020**, 41, 1006-1016.
5. K. K. Patra, Z. Liu, H. Lee, S. Hong, H. Song, H. G. Abbas, Y. Kwon, S. Ringe, J. H. Oh, *ACS Catal.* **2022**, 12, 10973-10983.

Line 28-30, Page 5; Line 1-6, Page 6 in the revised main article.

Comments 12: Please supplement the result analysis for Figure S5.

Reply 12: As shown in Supplementary Fig. 5, there are no crystallized copper species in the form of clusters or small particles in the structure.

Line 9-11, Page 5 in the revised main article.

Comments 13: “Also, Ce M_{4,5} absorption edge soft XAS spectra (Figure S9a) and O K-edge XANES spectra (Figure S9b) for CeO₂ and Cu-CeO₂ show no obvious difference” Please provide further explanation.

Reply 13: Thank you for your suggestions. The XANES at Ce M_{5,4}-edge of CeO₂ based materials correlates with the Ce 3d_{3/2} and 3d_{5/2} core level transitions into the 4f unoccupied electronic state, therefore it manifests the occupancy of the 4f orbital and valence states (Ce⁴⁺ and Ce³⁺). (Ref. 1) XANES spectra at the Ce M_{5,4}-edge were normalized and are shown in Supplementary Fig. 10a. The Ce M₄-edge is regarded as the replica of the Ce M₅-edge. Therefore, only M₅-edge is discussed herein. The intense peak S and peak P represent the tetravalent Ce (4f⁰) while the weak peak R indicates the contribution of trivalent Ce (4f¹) states (Ref. 2-3). As it vividly shows, Cu-CeO₂ sample exhibits a reduction of Ce³⁺ (peak R) compared with parent CeO₂ while an enhancement of Ce⁴⁺ (peak S). In other words, the Ce³⁺/(Ce³⁺ + Ce⁴⁺) ratio of Cu-CeO₂ is lower than that of CeO₂. The result is in good agreement with XPS results (Figure 2c, Supplementary Table 1). XPS measurements are mainly employed to investigate occupied orbital information and are more sensitive to valence states, while XAS detects unoccupied orbital information. It is obvious that the Ce³⁺ content for Cu-CeO₂ is nearly 18 % from XPS analysis, which is lower than that of CeO₂ (23 %). These results indicate that the introduction of Cu element into parent CeO₂ would reduce its surface Ce³⁺ concentration.

Supplementary Fig. 10b shows the O K-edge. Three main peaks are attributed to the O 2p states that are hybridized with Ce 4f, 5d(e_g) and 5d(t_{2g}) states, respectively. The intensity variation is attributed to the structural disorders induced by Cu doping owing to the formation of Cu-O-Ce coordination network (Ref. 1).

Supplementary Fig. 10(a). Soft XAS spectra of Ce $M_{4,5}$ absorption edges for CeO_2 and Cu-CeO_2 samples (left) and expanded view of shadow areas (right).

Figure 2c. Ce 3d photoelectron profiles of the CeO_2 and Cu-CeO_2 catalysts.

Supplementary Table 1. The area ratios between Ce^{3+} and Ce^{4+} species of different samples in Figure 2c.

Sample	$\text{Ce}^{3+}/(\text{Ce}^{3+}+\text{Ce}^{4+})$
CeO_2	0.23
Cu-CeO_2	0.18

Reference:

1. A. Sharma, M. Varshney, J. Park, *et al. Phys. Chem. Chem. Phys.*, **2015**, 17, 30065-30075.
2. L. Tao, Y. L. Shi, Y. C. Huang, *et al. Nano Energy*, **2018**, 53, 604-612.
3. P. Kumar, F. Chand, K. Asokan, *AIP Conf. Proc.*, **2019**, 2115, 030529.

This has been revised in manuscript: Line 7-17, Page 8.

Supplementary Fig. 9 in the revised Supplementary Information.

Comments 14: The Cu-CeO₂ air catalyst refers to the product obtained after heating the composite in air? Please explain or draw the synthesis steps and the obtained product in Figure 1a.

Reply 14: In the synthesis section in manuscript, in order to acquire the target product Cu-CeO₂ SSE, Cu-CeO₂ precursor was first calcined in air to get Cu-CeO₂ air sample, and then annealed under a H₂/Ar atmosphere.

Figure 1a. Schematic illustration of the synthetic procedure of Cu-CeO₂ sample.

This information has been added in Figure 1a in manuscript.

Comments 15: Figure S11b, in the steady-state kinetic assay, the substrate concentration establishment is not reasonable. Usually, a suitable substrate concentration range should straddle the K_m value and range roughly from 0.5 × K_m to 5 × K_m.

Reply 15: Thank you for pointing out this problem. We have adjusted the concentration of H₂O₂ in the steady-state kinetic assay. As described in the manuscript, “For steady-state kinetic assays, the system contains 50 μg/mL Cu-CeO₂, 208 μM TMB with 15-150 mM H₂O₂.”, and the K_m of Cu-CeO₂ was 30.76 mM. The results are shown in Supplementary Table 3.

Line 17-18, Page 23 in the revised main article.

Supplementary Table 3 in Supplementary Information.

Comments 16: In Figure S12 (Relative cyclic stability of CeO₂ and Cu-CeO₂), there is only a group of data.

Reply 16: We are sorry for our mistake. We have corrected the legend of Supplementary Fig. 13 as “Relative cyclic stability of Cu-CeO₂”.

This has been revised in Supplementary Fig. 13 in Supplementary Information.

For Reviewer #2:

Reviewer #2: The manuscript by Jiang et al. presents a systematic study on improved peroxidase-like activity and hydroxyl radical antioxidant capacity of Cu-CeO₂ nanozymes with respect to CeO₂ counterpart. The material characterization is quite extensive, but raises a few concerns as listed below. The DFT-based mechanistic study provides important insights into observed activity enhancements. Yet simulations consider only single-atom Cu species as active sites, while larger Cu species (clusters and nanoparticles) can also be present in the synthesized system. Possible contribution of those metal sites is not discussed. Overall I recommend this work for publication after resolving the questions below:

Thank you for your recognition of our manuscript. These constructive suggestions will help us shed light on the nature of the structure of Cu-CeO₂ and the response mechanism of antibacterial application. Acting on your recommendation, we have done an adequate investigation and analysis. The related explanations are provided below.

Comments 1: The title reads “Ceric Dioxide”. As there is no “ceric” element in periodic table, the title must be updated with “cerium dioxide” or “ceria”.

Reply 1: Thanks for your kind reminders. The “Ceric Dioxide” in the title has been replaced by “Ceria”.

This has been revised in the manuscript and Supplementary Information.

Comments 2: STEM EDX imaging with such a large pixel size (in the range of few nm) as shown in Fig S4 cannot provide sufficient evidence for atomically dispersed nature of copper species. Effectively these data only suggests that Cu is homogenously distributed over the sample on the scale of few nm, but not necessarily on the atomic level. The EXAFS data indeed suggests that Cu species are highly-dispersed, but does not rule out the presence of Cu clusters and/or nanoparticles of sub-nm to few-nm size. Recent work (10.1063/5.0008748) demonstrates that amorphous CuO nanoparticles below 3 nm can exhibit an EXAFS spectrum virtually resembling atomically dispersed CuO species. Thus EXAFS and STEM EDX (without atomic-resolution EELS/EDX elemental mapping) are not sufficient to unambiguously conclude the single-atom nature of metallic sites. Authors should consider use of CO-probe molecule FTIR studies to confirm single-atom nature of metal sites. Otherwise, the authors should refrain from claiming that all Cu species are atomically dispersed or represented by single sites.

Reply 2: Thanks for your suggestions. Based on the clarification in this article, the second shell of M-M peaks may be significantly suppressed owing to the ultra-low intensity in response to disordered effect (Ref. 1). Thus we performed CO-probe molecule FTIR measurements to

investigate the nature of Cu metal sites. In situ DRIFTS was used to investigate the adsorption ability of Cu-CeO₂. As shown in Supplementary Fig. 9, Cu-CeO₂ catalyst exhibits a band around 2105 cm⁻¹, assigned to the linear CO adsorbed on Cu⁺ sites (Cu⁺-CO), indicating that CO was adsorbed on the Cu⁺ sites (Ref. 2-6). Additionally, the adsorption intensity of the peak gradually increased with the time, and reached saturation adsorption at 480s with a maximum adsorption peak at 2111cm⁻¹. Traditionally, the IR band at 2069 cm⁻¹ is regarded as the CO adsorption on Cu⁰ site (Ref. 2, 3). This indicates that there are Cu⁺ species in this Cu-CeO₂ catalyst. Besides, XANES profiles in Figure 2d and FT-EXAFS spectra in Figure 2e also indicate that there is no formation of Cu-Cu bond. Based on the steady-state kinetic results of POD-like activities of Cu-CeO₂ samples with different Cu contents in Supplementary Table 3, when the Cu content increased from 2% to 5%, the V_{max} increased by ~2.13 folds, and the turnover rate showed a slight decrease, indicating the potential existence of small amount of Cu nano-clusters in our Cu-CeO₂ sample.

Supplementary Fig. 9. In situ DRIFTS study of CO adsorption on Cu-CeO₂. (2% CO/Ar flow rate, 30 mL min⁻¹; catalyst mass, 100 mg; temperature, 30 °C)

Reference:

1. K. Feng, H. Z. Zhang, J. Gao, J. B. Xu, Y. M. Dong, Z. H. Kang, J. Zhong, *Appl. Phys. Lett.* **2020**, 116, 191903.
2. L. Q. Kang, B. L. Wang, A. T. Guntner, *et al. Angew. Chem. Int. Edit.* **2021**, 60, 14420-14428.
3. H. X. Liu, S. Q. Li, W. W. Wang, W. Z. Yu, W. J. Zhang, C. Ma, C. J. Jia, *Nat. Commun.*, **2022**, 13, 867.
4. W. W. Wang, W. Z. Yu, P. P. Du, H. Xu, Z. Jin, R. Si, C. Ma, S. Shi, C. J. Jia, C. H. Yan, *ACS Catal.*, **2017**, 7, 1313-1329.
5. W. Z. Yu, W. W. Wang, S. Q. Li, X. P. Fu, X. Wang, K. Wu, R. Si, C. Ma, C. J. Jia, C. H. Yan, *J. Am. Chem. Soc.*, **2019**, 141, 17548-17557.
6. A. L. Chen, X. J. Yu, Y. Zhou, *et al. Nat. Catal.*, **2019**, 2, 334-341.

This has been revised in manuscript: Line 15-16, Page 7; Line 1-6, Page 8; Line 9-17, Page 16.
Supplementary Fig. 9 in Supplementary Information.

Comments 3: The authors performed XPS study of Ce state in both CeO₂ and Cu-CeO₂ samples. I would like to complement the authors on successful efforts in fitting complex Ce 3d spectra. For completeness, I would recommend to refer to the literature which describes the used fitting model. As the authors suggest on the basis of Raman data that addition of Cu influences the concentration of defects/oxygen vacancies (and thus Ce³⁺ concentration), it is highly recommended to report the actual Ce³⁺/Ce⁴⁺ ratios derived from the Ce 3d fitting shown in Fig 2c. From the visual assessment I can see that the intensity of Ce³⁺ peaks in CeO₂ is higher than in Cu-CeO₂, which would contradict the claim on concentration of defect based on Raman data. Could the authors provide the actual numbers and discuss these data quantitatively?

Reply 3: Thanks for your reminders, these questions would help us gain further insight into the nature of the structure of Cu-CeO₂ sample. The XPS spectra of Ce 3d (Figure 2c) are fitted into 10 peaks, which are attributed to the Ce⁴⁺ species at v, v'', v''', u, u'', u''' and the Ce³⁺ species at v₀, v', u₀, u', respectively. (Ref. 1-4) Supplementary Table 1 shows the actual Ce³⁺/Ce ratios of both CeO₂ and Cu-CeO₂ samples. It is obvious that the Ce³⁺ content for Cu-CeO₂ is nearly 18 % from XPS analysis, which is lower than that of CeO₂ (23 %). These results indicate that the introduction of Cu element into parent CeO₂ would reduce its surface Ce³⁺ concentration. Similar phenomenon has also been reported in previous report (Ref. 4). According to the authors' calculations and structural characterization, the replacement of one Ce³⁺ next to one oxygen vacancy (V_O) by a Cu²⁺ should be viewed as being accompanied by a conversion of the other Ce³⁺ to Ce⁴⁺, and thus, the increase of Cu²⁺ doping level will lead to a decrease of Ce³⁺/Ce⁴⁺ ratio from the original undoped CeO₂ for the charge balance. The O 1s spectra of CeO₂ and Cu-CeO₂ samples are illustrated in Supplementary Fig. 8b. The peak appearing at approximately 529 eV (O_I) can be attributed to the lattice oxygen of Ce⁴⁺ and the feature at 530.7 eV (O_{II}) corresponds to oxygen vacancies or the lower-coordination lattice oxygen of Ce³⁺. (Ref. 1, 2) Also, the higher binding energy feature at 531.6 eV is also associated with the presence of surface hydroxy-containing groups. It is obvious that parent CeO₂ possesses a larger O_{II}/(O_I + O_{II}) ratio (23.3 %) than that of Cu-CeO₂ (19.8 %), which is in line with the Ce 3d XPS results.

For proton exchange membrane fuel cells (PEMFCs), the membrane always suffers from severe chemical degradation from active radical attack. Since it can react with free radicals, ceria has been regarded as one of the most promising additives to quench •OH in fuel cells (Ref. 5). The results indicate that the hydroxyl radical scavenging is significantly reduced without an adequate amount of Ce³⁺ in the membrane (Ref. 6). In other words, the free radical scavenging property of ceria is closely related to Ce³⁺ content. Generally speaking, the higher the ratio of Ce³⁺/Ce⁴⁺ for CeO₂-based material, the stronger the hydroxyl radical antioxidant capacity (HORAC) (Ref. 7-9). In other words, compared with parent CeO₂, a lower Ce³⁺ concentration of our Cu-CeO₂ sample demonstrates an excellent inhibitory effect against the scavenge of oxygen free radicals. The decrease of free radicals scavenging efficiency will maximize its impact to achieve a good antibacterial property.

Raman spectra are usually used to detect the defect sites in CeO₂ samples. When CeO₂ is synthesized, Ce³⁺ and oxygen vacancies (O_v) are created on the surface. Generally speaking, the band centered at ~460 cm⁻¹ is characteristic of the triply degenerate Raman active mode of cubic

fluorite structure of ceria (F_{2g}) while the bands at 500 to 600 cm^{-1} are assigned to a defect-induced mode (D). The intensity of these two spectra is different, and we have made modification in Figure 2b. The softening of the F_{2g} mode is accompanied by the appearance of a broad feature centered around 600 cm^{-1} . (Ref. 10) The Raman mode is extremely sensitive to changes in the oxygen stoichiometry around the Ce atom. For pure CeO_2 , the D mode vibration is produced by the oxygen vacancy caused by the presence of Ce^{3+} . However, for doped CeO_2 , it is misleading to simply refer to the peak at ca. 600 cm^{-1} as a response signal to O_v . In addition to oxygen vacancies, point defects and the increasing ordering level can also contribute to the variation of D band (Ref. 11, 12). As pointed out by a special study which focused on doped CeO_2 , the appearance of D band is due to the formation of defect species in Ce-O coordination, which leads a consequence that the vibration signal of Ce-O cannot be cancelled in all directions. (Ref. 11) The defects can be divided into an α defect at 560 cm^{-1} and a β defect at 600 cm^{-1} . The former band was assigned to defect species including an oxygen vacancy, while the latter was assigned to the formation of the MO_8 -type complex but did not contain any oxygen vacancies. For β defect, the oxygen vacancy concentration is constant while the increasing disordering level of the sample can also lead to D vibrational modes. Thus the weak peak at about 600 cm^{-1} in the Raman spectrum of our Cu- CeO_2 sample can be attributed to the β -peak signal mentioned in the above literature. Since there exists a Cu-O-Ce coordination structure in Cu- CeO_2 catalyst, the interaction between Cu-O and Ce-O is not equal, which also leads to the difference in the vibration of Ce-O in different directions. It's reported by Huang and his coworkers that the isolated Cu ions tend to preferentially deposit at the defective sites of CeO_2 , likely the surface oxygen vacancies (Ref. 13). Indeed, Oh illustrates that atomically dispersed copper atoms would cause the content of surface V_O and Ce^{3+} to decrease, which is also consistent with our XPS results. (Ref. 1) Overall, the presence of the D mode cannot be used as direct evidence for the increased concentration of oxygen vacancy.

Figure 2c. Ce 3d photoelectron profiles of the CeO_2 and Cu- CeO_2 catalysts.

Supplementary Fig. 8b. O 1s photoelectron profiles of the CeO₂ and Cu-CeO₂ catalysts.

Supplementary Table 1. The area ratios between Ce³⁺ and Ce⁴⁺ species of different samples in Figure 2c.

Sample	Ce ³⁺ /(Ce ³⁺ +Ce ⁴⁺)
CeO ₂	0.23
Cu-CeO ₂	0.18

Figure 2b. Raman spectra of the CeO₂ and Cu-CeO₂ catalysts.

Reference:

1. K. K. Patra, Z. Liu, H. Lee, *et al. ACS Catal.* **2022**, 12, 10973-10983.
2. S. C. Rood, O. P. Algaba, A. T. Princep, *et al. Chem. Eur. J.*, **2021**, 27, 2165-2174.
3. D. Jiang, Y. G. Yao, T. Y. Li, *et al. Angew. Chem. Int. Edit.*, **2021**, 60, 26054-26062.
4. Y. F. Wang, Z. Chen, P. Han, *et al. ACS Catal.* **2018**, 8, 7113-7119.
5. Z. Y. Rui, Q. B. Li, Y. X. Hou, *et al. RSC Adv.*, **2021**, 11, 32012-32021.

6. K. H. Wong, E. Kjeang. *J. Electrochem. Soc.*, **2017**, 164, F1179-F1186.
7. Y. F. Fan, P. Y. Li, B. B. Hu, *et al. Inorg. Chem.*, **2019**, 58, 7295-7302.
8. I. Celado, M. De Nicola, C. Mandoli, *et al. ACS Nano*, **2011**, 5, 4537-4549.
9. Y. Xue, Q. F. Luan, D. Yang, *et al. J. Phys. Chem. C*, **2011**, 115, 4433-4438.
10. J. S. Elias, N. Artrith, M. Bugnet, *et al. ACS Catal.* **2016**, 6, 1675-1679.
11. L. Li, F. Chen, J. Q. Lu, *et al. J. Phys. Chem. A*, **2011**, 115, 7972-7977.
12. M. Sarkar, M. R. Rajkumar, S. Tripathy, *et al. Mater. Res. Bull.*, **2012**, 47, 4340-4346.
13. Y. X. Gao, Z. H. Zhang, Z. R. Li, *et al. Chin. J. Catal.*, **2020**, 41 1006-1016.

This has been revised in manuscript: Line 28-30, Page 5; Line 1-25, Page 6; Figure 2b, 2c. Supplementary Fig. 8b, Supplementary Table 1 in Supplementary Information.

Comments 4: The much higher ROS generation activity of Cu-CeO₂ with respect to pure CeO₂ shown in Fig 3e is striking. As for some test the reaction media is acidic, is it possible that Cu leaches away into solution and acts as a homogeneous catalyst? Had ICP or any other analytical method been used to conclude on possible Cu dissolution? Could the authors perform a control measurement with pure CuO under identical experimental conditions (Fig 4c)? Please also fix a typo in Fig 4d - I assume it is H₂O₂ instead of H₂O in EPR data labelling.

Reply 4: Thank you for your helpful suggestion. We have supplemented the ICP-MS result of Cu-CeO₂ after cyclic POD-like reactions in phosphate buffer saline (pH=7.4, the same buffer as that of ROS generation and antibacterial experiments). As described in the manuscript, “The ICP-MS analysis showed that Cu accounts for 4.43 wt% of the Cu-CeO₂ sample. Hence, when the concentration of Cu-CeO₂ reaches 200 µg/mL, the total content of Cu is 8.858 µg/mL. The content of Cu in the supernatant after cyclic reaction was below 0.500 µg/mL, much lower than the total amount of Cu in the reaction system.” As a result, we speculate that Cu leakage in ROS generation and antibacterial experiments is minimal.

In addition, we have supplemented the POD-like activity test of pure CuO. As shown in Supplementary Fig. 12 and Supplementary Table 3, Cu-CeO₂ SSE exhibited significantly enhanced POD-like activity compared to CuO. We have also corrected the annotations of H₂O₂ in all figures.

Supplementary Fig. 12. Steady-state kinetic assay of POD-like catalytic activity of (a) CeO_2 , $Cu-CeO_2$, and (b) CuO , 2%, and 10% $Cu-CeO_2$. Michaelis-Menten plot with different concentrations of H_2O_2 .

This has been revised in manuscript: Line 6-7, Line 20-23, Page 9.

Supplementary Fig. 12, Supplementary Table 3 in Supplementary Information.

Comments 5: The DFT modelling and mechanistic study solely rely on single-atom structure of Cu species in prepared nanozymes. As discussed above, from the presented data it is difficult to conclude that all Cu species are atomically dispersed. In reality, clusters and nanoparticles of transition metals often accompany single-atom species in the sample. It would be worthwhile if authors could provide some calculations (see e.g., 10.1039/D1SC01201K) or at least discuss a possible contribution of small CuO clusters to the POD-like and HORAC activity.

Reply 5: To elucidate the possible influence of small CuO cluster on the surface, additional DFT calculations were performed. According to previous literatures, small Cu nano-clusters on CeO_2 facets prefer to be in the form of monolayers, a $(CuO)_3-CeO_2$ model was thus established (Ref. 1-2). The stable planar $(CuO)_3$ cluster with similar symmetry of CeO_2 (111) surface was loaded. Considering the aqueous environment of the actual experimental condition, water molecule was added and found dissociated spontaneously on the top of the CuO cluster under the strong influence of 3 under coordinated Cu , forming the hydrated site as shown in Supplementary Fig. 25-26. Due to the structural complexity of the cluster, there are two possible reaction sites, i.e., the Cu_3 site and the Cu_2Ce site which surround the $*OH$ intermediate. Both HORAC and POD-like mechanisms were calculated on these sites. The results indicated that for POD-like pathway, the energy consumption of the PDS are sharply reduced to 0.796 eV (Cu_3) and 0.915 eV (Cu_2Ce) mainly due to the strong interaction of the intermediates ($*OH$) with multiple coordination atoms. Thus, $(CuO)_3$ cluster boosted the activity of $\cdot OH$ radical generation. While for the HORAC pathway, the strong binding of O_2 with multiple atoms further increase the energy consumption of the PDS (O_2 desorption) to 1.081 eV (Cu_3) and 1.015 eV (Cu_2Ce), which deactivated the HORAC process. The above results suggest that small Cu nano-clusters also possess oxidant activities. Based on the steady-state kinetic results of POD-like activities of $Cu-CeO_2$ samples with different Cu contents in Supplementary Table 3, when the Cu content increased from 2% to 5%, the V_{max}

increased by ~ 2.13 folds, and the turnover rate showed a slight decrease, indicating that potential existence of small amount of Cu nano-clusters in Cu-CeO₂ sample may contribute to the overall POD-like activity of the nanozyme. However, when the Cu content continued to increase to 10%, the V_{\max} did not increase correspondingly, but remained comparable to that of the 5% sample. Meanwhile, the turnover rate drastically decreased by ~ 2.10 folds, suggesting that the presence of a large number of Cu nano-clusters may actually inhibit the overall catalytic activity of the nanozyme. The phenomenon we observed is consistent with that reported in previous literature (Ref. 3).

Supplementary Fig. 25. The DFT optimized geometry of reaction intermediates on Cu₃ site. (a) clean slab, (b) *H₂O₂, (c) *OH, (d) *H₂O, (e) *H₂O + *O, (f) *O, (g) *O₂, (h) *H₂O + *O₂.

Supplementary Fig. 26. The DFT optimized geometry of reaction intermediates on Cu_2Ce site. (a) clean slab, (b) $^*\text{H}_2\text{O}_2$, (c) $^*\text{OH}$, (d) $^*\text{H}_2\text{O}$, (e) $^*\text{H}_2\text{O} + ^*\text{O}$, (f) $^*\text{O}$, (g) $^*\text{O}_2$, (h) $^*\text{H}_2\text{O} + ^*\text{O}_2$.

Reference:

1. Y. Q. Su, G. J. Xia, Y. Y. Qin, et al. *Chem. Sci.*, **2021**, 12, 8260.
2. C. H. Zheng, H. F. Bu, F. Yang, et al. *Energy Technol.*, **2022**, 10, 2100161.
3. J. W. Xu, X. L. Zheng, Z. P. Feng, et al. *Nat. Sustain.*, **2021**, 4, 233.

This has been revised in manuscript: Line 23-29, Page 15; Line 1-18, Page 16.

Supplementary Fig. 25-26 in the revised Supplementary Information.

Comments 6: The discussion of in vitro experiments is too technical and contains a lot of experimental details (e.g., concentrations) which should be listed in the methods section. It would improve readability if authors would focus on the main findings of these tests.

Reply 6: We must thank you again for your valuable suggestions. We have optimized the description of the in vitro experiment section, deleted some methodological content, and added discussion on experimental findings, as discussed in the manuscript: For *E. coli*, due to its high

sensitivity to H₂O₂, the colony number showed a decreasing tendency with the increase of H₂O₂ concentration. Notably the antibacterial effect gradually deteriorated with the increase of CeO₂ concentration, which was even worse than that of H₂O₂ with the same concentration. Cu-CeO₂ + H₂O₂ group still showed a positive correlation between antibacterial effect and nanozyme and H₂O₂ concentration. The phenomenon observed in our study is similar to that reported by Zhu *et al* (Ref. 1). According to their report, the synthesized spherical CeO₂ nanozyme with a particle size of ~150 nm exhibited optimistic POD-like activity at pH = 4.0-6.0. However, the bactericidal effect of CeO₂ + H₂O₂ on *E. coli* in PBS (pH = 4.0) was significantly weaker than that of using H₂O₂ alone, and no significant ·OH was detected in the CeO₂ + H₂O₂ system. Therefore, they speculated that the ROS scavenging ability of CeO₂ hinders the decomposition of H₂O₂ to generate ·OH, and its POD-like activity did not contribute to the antibacterial effect of the CeO₂ + H₂O₂ system.

Reference:

1. W. S. Zhu, L. Y. Wang, Q. S. Li, *et al. Molecules*, **2021**, 26, 3747.

This has been revised in manuscript: Line 4-11, Page 19.

For Reviewer #3:

Reviewer #3: This study developed a single-atom copper modified CeO₂ nanozyme with multiple catalytic activities and inhibited free radical scavenging capacity for antibacterial application. The catalytic mechanism regulated by the interaction between single-atom Cu and CeO₂ carrier was explored using experimental and theoretical methods. The antibacterial performance of Cu-CeO₂ systems in vitro and in vivo was also confirmed. However, I do not believe that the novelty and quality of this manuscript meet the high standard of Nature Communications.

We really appreciate that you took time out from your busy schedule to review our manuscript and share your concerns. These constructive comments help us to improve and complete our demonstration. Despite achieved much progress in various catalysis (Table 1 below), the “star catalyst” Cu-CeO₂ has been rarely reported in antibacterial applications. The study systemically researched the efficient regulation of Cu single site on the direction of oxidant and antioxidant activities of ceria for the first time through experiments and DFT calculations, namely the conversion of POD and HORAC. The study demonstrates the great potential of this system for antibacterial treatment, and enriches the theory of metal-substrate interaction. This study also provides a new clue for regulating the complex activities of nanozymes. Apart from antibacterial therapy, our regulating strategies can also be applied to other applications related to free radical biology, such as tumor inhibition, sensing, degradation of organic pollutants, etc.

Additionally, based on the comments of reviewer 1 and reviewer 2, we also have done a large amount of experiments and made adequate analyses and modifications. The quality of the manuscript has been improved. We hope the current revised manuscript could fall in the high impact of the Nature Communications. The related explanations are described below.

Table 1 Current reported Cu-CeO₂ based catalysts on various catalytic reactions.

Research Fields	Main Findings	Year	Reference
Electrochemical CO ₂ Reduction	The tunable CH ₄ /C ₂ H ₄ selectivity of Cu-CeO ₂ during the electrochemical CO ₂ RR by precisely controlling the structure and valence state of the Cu active sites on CeO ₂ ; the unique tunability is a result of the immiscible nature of the CuO and CeO ₂ binary oxide system	2023	1
Electrocatalytic CO ₂ Reduction	Cu ²⁺ in Cu-Ce-O _x can enhance the adsorption stability of the *CO intermediate, which in turn promotes its further hydrogenation and suppresses the dimerization, thereby significantly improving the selectivity for CH ₄ ; retaining high-valence-state metal catalytic active sites through material structure design	2022	2
Reverse Water Gas Shift (RWGS)	Highly dispersed active copper clusters with high loading (15 wt.%) exhibited excellent catalytic performance to catalyze the RWGS reaction at high operating temperature; The harsh reaction conditions of high temperature and reductive atmosphere caused the ceria support sintered partially, while the interaction between copper and ceria maintained well	2022	3
Electrocatalytic Urea Synthesis	The reconstitution of single atoms (Cu ₁) to clusters (Cu ₄) during electrolysis; the dynamic and reversible transformation of clusters to single-atom configurations occurs when the applied potential was switched to an open-circuit potential, yielding a catalyst with high structural and electrochemical stabilities.	2023	4
NO Reduction by CO	The abundant Cu ⁺ /Ce ³⁺ paired sites with surface synergetic oxygen vacancies (SSOV) on CuO _x -CeO ₂ catalyst could effectively facilitate the adsorption and activation of CO and NO	2023	5
CO Oxidation	A theoretical study on CO oxidation over substituted Cu ₁ /CeO ₂ single atom catalysts; dynamic evolution of metal-support coordination can significantly change the electronic structure of the active center; extending hemilability effects to single atom heterogenous catalysts	2023	6
Consecutive Hydrogenation	Proposed a low-temperature atomic diffusion strategy; stable Cu clusters on CeO ₂ exhibit high H ₂ dissociation ability and low adsorption ability towards intermediate products, resulting in the efficient semi-hydrogenation performance	2023	7

Antibacterial Application	Revealing the structure-activity relationship of Cu-CeO ₂ single site enzyme; precise and efficient regulation of redox reaction pathway and catalytic activity of CeO ₂ via anchoring Cu single site, exhibiting excellent antibacterial properties	-	This work
--	---	-----------

Reference:

1. S. W. Hong, H. G. Abbas, K. Jang, *et al. Adv. Mater.*, **2023**, 35, 2208996.
2. X. L. Zhou, J. Q. Shan, L. Chen, *et al. J. Am. Chem. Soc.*, **2022**, 144, 5, 2079-2084.
3. H. X. Liu, S. Q. Li, W. W. Wang, *et al. Nat. Commun.*, **2022**, 13, 867.
4. X. X. Wei, Y. Y. Liu, X. R. Zhu, *et al. Adv. Mater.*, **2023**, 35, 2300020.
5. W. Tan, Y. D. Cai, S. H. Xie, *et al. Chem. Eur. J.*, **2023**, 456, 140807.
6. Z. Chen, Z. Y. Liu, X. Xu. *Nat. Commun.*, **2023**, 14, 2512.
7. D. W. Yao, Y. Wang, Y. Li, *Nat. Commun.*, **2023**, 14, 1123.

Comments 1: CeO₂ nanosphere was prepared in this study. However, the illustrations of CeO₂, Cu-CeO₂ precursor and Cu-CeO₂ SASE in Figure 1a look like a hollow structure, which could readily confuse readers.

Reply 1: Thank you for your reminders. Actually, monodisperse CeO₂ mesoporous spheres have been prepared in this manuscript based on the morphology characterization results. This has been reported in previous study (Ref. 1). The related illustrations in Figure 1a have been revised.

Figure 1a. Schematic illustration of the synthetic procedure of Cu-CeO₂ sample.

Reference:

1. X. Liang, J. J. Xiao, B. H. Chen, *et al. Inorg. Chem.*, **2010**, 49, 8188-8190.

This information has been revised in Figure 1a in manuscript.

Comments 2: Multiple catalytic properties of Cu-CeO₂ were investigated in Figure 3. It was concluded that the higher CAT, SOD, as well as OXD-like activities of Cu-CeO₂ may also contribute to reproducing H₂O₂ through cascade reactions. Does it mean that the consumption of H₂O₂ by Cu-CeO₂ is slower than that of CeO₂? The consumption of H₂O₂ by Cu-CeO₂ and CeO₂ should be provided?

Reply 2: Thank you for your rigorous consideration. We have supplemented the experiment of H₂O₂ consumption in CeO₂ + H₂O₂ and Cu-CeO₂ + H₂O₂ systems. As shown in Supplementary Fig. 15, as the reaction time prolongs, both systems showed a trend of H₂O₂ consumption, and Cu-CeO₂ maintains a higher consumption than CeO₂ during long-term reaction process. This may be caused by the enhanced multiple catalytic activities of Cu-CeO₂, which accelerate the multiple pathway conversion related to H₂O₂.

Supplementary Fig. 15. H₂O₂ consumption of CeO₂ and Cu-CeO₂. (***)*P* < 0.001)

This has been revised in manuscript: Line 17-22, Page 11.
 Supplementary Fig. 15 in Supplementary Information.

Comments 3: On Line 203, “the fluorescence intensity of CeO₂ group drastically decreased by ~94.31%, while Cu-CeO₂ group merely decreased by ~1.92% (Figure S14)”. Figure S14 shows relative clearance rate of ·OH based on the result in Figure 3d. However, the fluorescence intensity of Cu-CeO₂ group was even higher than that of H₂O group. Please explain it and provide the calculating formula of relative clearance rate of ·OH.

Reply 3: We have modified the explanation of the HORAC activity results, and added the as described in the manuscript: “within 10 min of reaction, the fluorescence intensity of CeO₂ group drastically decreased by ~94.31%, compared with baseline, while Cu-CeO₂ group only decreased by ~1.92%.” The calculating method was also added, as described in the manuscript: “Immediately after the reaction, the fluorescence intensity was measured with a multimode plate reader ($\lambda_{em} = 490$ nm, $\lambda_{ex} = 525$ nm) and recorded every minute, and the final fluorescence intensity was compared with the initial fluorescence intensity (baseline) of each group. The relative change of fluorescence intensity was calculated as follow:

$$\text{Relative change} = |F_0 - F_t| / F_0 \times 100\%$$

Where F_0 represents the initial fluorescence intensity (baseline), and F_t represents the fluorescence intensity after certain timen of reaction.

In addition, we measured the changes of ·OH species quantitatively through EPR. As shown in Supplementary Fig. 16 and Supplementary Table 4, after 10 minutes of reaction, both FeCl₂ and

FeCl₂ + Cu-CeO₂ group exhibited typical DMPO-OH signals. The signal intensity of FeCl₂ + Cu-CeO₂ group was higher than that of FeCl₂ group, while the signal of FeCl₂+CeO₂ group was almost invisible, which is consistent with the pattern of the fluorescence results. Quantitative calculation showed that the ·OH scavenging rate of CeO₂ was 97.26 % at 10 minutes, while the spin concentration of DMPO-OH in FeCl₂ + Cu-CeO₂ group reached 158.62 % of that in FeCl₂ group.

Supplementary Fig. 16. EPR spectrum of HORAC reactions of CeO₂ and Cu-CeO₂.

Supplementary Table 4. Quantitive countings of ERP spins of HORAC reactions of CeO₂ and Cu-CeO₂.

Group	Spins	Spin concentration (M)
FeCl ₂	3.91×10^{13}	3.64×10^{-6}
FeCl ₂ +CeO ₂	1.07×10^{12}	9.98×10^{-8}
FeCl ₂ +Cu-CeO ₂	6.20×10^{13}	5.77×10^{-8}

This has been discussed in manuscript: Line 27-29, Page 11; Line 1-11, Page 12; Line 7-13, Page 24.

Supplementary Fig. 16, Supplementary Table 4 in Supplementary Information.

Comments 4: Only ·OH generation was confirmed by EPR characterization. What about O₂^{·-}?

Reply 4: Thank you for your reminders. We have supplemented the O₂^{·-} generation experiment using EPR, as shown in Supplementary Fig. 18, only Cu-CeO₂ group showed significant DPMO-O₂^{·-} signal.

Supplementary Fig. 18. EPR spectrum of OXD reactions of CeO₂ and Cu-CeO₂.

This has been discussed in manuscript: Line 5-7, Page 13.

Supplementary Fig. 18 in the revised Supplementary Information.

Comments 5: It is still not clear that what the valence state of Cu is. The statements, “the isolated single Cu atoms bear a high oxidation valence state in Cu-CeO₂ sample” and “Cu atoms are present as a monodispersed state in Cu-CeO₂ sample”, seem to be inconsistent.

Reply 5: Thank you for your reminders. Since Cu is coordinated with oxygen in ceria substrate, the resulting Cu-O-Ce coordination structure means the valence state of Cu is not zero. Our previous works have illustrated that the active metal sites in single atom catalyst, which coordinated with heteroatoms (O, N, P, S etc.), exhibited high valence states (Ref. 1-3). The statement “Cu atoms are present as a monodispersed state in Cu-CeO₂ sample” means the dispersive feature of Cu species in Cu-CeO₂ sample.

Reference:

1. Z. D. Zhang, J. Wang, X. H. Ge, *et al. J. Am. Chem. Soc.*, **2023**, 145, 22836-22844.
2. Y. J. Chen, B. Jiang, H. G. Hao, *et al. Angew. Chem. Int. Ed.*, **2023**, 62, e202301879.
3. X. H. Sun, Y. X. Tuo, C. L. Ye, *et al. Angew. Chem. Int. Ed.* **2021**, 60, 23614-23618.

This has been discussed in manuscript: Line 10-11, Page 7.

Comments 6: Relative cell viability of hGF and hPDLSc was investigated after treatment with different concentrations of Cu-CeO₂. However, Cu-CeO₂ + H₂O₂ system was used for the in vivo antibacterial and wound healing experiments. Generally, the level of ROS is high in the infected wounds, which delays the wound healing process. Therefore, antioxidant agents have attracted

much attention in the field of wound dressing. Please indicate the advantage of Cu-CeO₂ + H₂O₂ system for infected wound healing.

Reply 6: We must thank you again for your valuable suggestions and rigorous opinions. We have supplemented discussion of the advantage of Cu-CeO₂ for infected wound healing, as discussed in the manuscript: “On the other hand, CeO₂, as an excellent antioxidant nanozyme, has received widespread attention in promoting injury healing (Ref. 1-2). In this study, although the ·OH scavenging capacity of Cu-CeO₂ was selectively inhibited, the scavenging activities of common ROS (i.e., H₂O₂ and O₂^{·-}) in the injury microenvironment were enhanced compared to CeO₂. This may be one of the advantages of Cu-CeO₂ in treating infected wounds. It is clear that within the scope of this study, Cu-CeO₂ plays an important role in the early anti-infectious process. However, its ability to accelerate wound healing in the subsequent antioxidant and tissue healing processes has not been thoroughly studied, which is also a limitation of this study.”

Reference:

1. M. Z. Zhang, X. Y. Zhai, T. F. Ma, *et al. ACS Nano*, **2023**, 17, 4433-4444.
2. X. T. Zhou, M. You, F. H. Wang, *et al. Adv. Mater.*, **2021**, 33, 2100556.

This has been discussed in manuscript: Line 8-15, Page 22.

Reviewers' Comments:

Reviewer #1:

Remarks to the Author:

The manuscript has been significantly improved. However, there are still some issues should be addressed before publication in Nature Communications.

1. In Fig 9, the authors performed CO-probe molecule FTIR measurements to investigate the nature of Cu metal sites. They assigned the band around 2105 cm^{-1} to the linear CO adsorbed on Cu^+ sites. Can the CeO_2 adsorb CO? Some results should be provided.
2. The XPS result of Cu in Cu- CeO_2 should be provided to certify the Cu^+ in Cu- CeO_2 .
3. Why does Cu^{2+} adsorbed and deposited on ceria become Cu^+ after calcined in air?
4. The author should explain why the introduction of Cu element into parent CeO_2 reduce its surface Ce^{3+} concentration.
5. Is the decrease of HORAC activity and the increase of POD, OXD, CAT, and SOD-like activities of Cu- CeO_2 due to the change in the valence state of Ce, Cu acting as a monoatomic catalyst, or the interaction between the two? Is there any data to support this?
6. How about the pH of mouse skin wound infection? Why can the Cu- CeO_2 catalyze the ROS generation for antibacterial performance and ROS scavenging for antioxidant simultaneously? It seems two opposite performances. The authors should clearly explain these aspects.

Reviewer #2:

Remarks to the Author:

The authors extensively revised the manuscript and performed additional experiments and calculations which strengthened their earlier findings. I support publication of this manuscript.

Reviewer #3:

Remarks to the Author:

The authors have revised their manuscript according to the previous reviewers' comments, and the quality of the manuscript has been improved. I believe now the manuscript can be considered for publication in the current form.

For Reviewer #1:

Reviewer #1: The manuscript has been significantly improved. However, there are still some issues should be addressed before publication in Nature Communications.

Thank you for your affirmation of our manuscript. These valuable and professional suggestions are very helpful for us to improve our manuscript. We have already conducted sufficient experiments and made detailed clarifications and modifications based on your comments.

Comments 1: In Fig 9, the authors performed CO-probe molecule FTIR measurements to investigate the nature of Cu metal sites. They assigned the band around 2105 cm^{-1} to the linear CO adsorbed on Cu^+ sites. Can the CeO_2 adsorb CO? Some results should be provided.

Reply 1: Thanks for your suggestions. We have supplemented the FTIR experiment of parent CeO_2 under the same testing conditions as control. As shown in Supplementary Fig. 9b, the two obvious bands are characteristic for CO bound to ceria surfaces, namely the gaseous peak of CO. (Ref. 1) It should be noted that the IR intensity of the Cu-CO bands is much higher than that of the ceria-related ones owing to the different types of interaction of CO with copper and ceria. The bonding of CO to the defect site on ceria is dominated by the weak electrostatic interaction (Stark effect), while CO adsorbs more strongly on electron-rich Cu species via π back-donation. It means that the CO absorption on parent CeO_2 is absent.

Reference:

1. A. L. Chen, X. J. Yu, Y. Zhou, et al. *Nat. Catal.*, **2019**, 2, 334-341.

Supplementary Fig. 9b. In situ DRIFTS study of CO adsorption on CeO_2 sample.

This has been revised in manuscript: Line 3, Line 9-10, Page 8.

Supplementary Fig. 9b in Supplementary Information.

Comments 2: The XPS result of Cu in Cu-CeO₂ should be provided to certify the Cu⁺ in Cu-CeO₂.

Reply 2: Thanks for your reminders. Cu 2p XPS pattern of Cu-CeO₂ is shown in Figure R1. According to the previous reports (Ref. 1-2), the XPS peaks centered at 933.6 and 932.5 eV can be attributed to the Cu 2p_{3/2} region, corresponding to Cu(II) and Cu(I)/Cu(0), respectively. For CeO₂, the surface oxygen is prone to be removed in a vacuum condition, resulting in the reduction of Cu species coordinated with oxygen. Therefore, the valence information obtained under the high vacuum condition of XPS may not be accurate, as the surface Cu²⁺ species were reduced to Cu⁺ or Cu⁰. In contrast, the synchrotron radiation test is carried out under atmospheric pressure, which can more accurately reflect the valence and structural information of Cu in catalyst.

Fig. R1. Cu 2p_{3/2} XPS pattern of Cu-CeO₂.

Reference:

1. W. W. Wang, W. Z. Yu, P. P. Du, *et al. ACS Catal.*, **2017**, 7, 1313-1329.
2. H. X. Liu, S. Q. Li, W. W. Wang, *et al. Nat. Commun.*, **2022**, 13, 867.

Line 7-9, Page 7 in the revised main article.

Comments 3: Why does Cu²⁺ adsorbed and deposited on ceria become Cu⁺ after calcined in air?

Reply 3: For CO-probe molecule FTIR tests, the measurements are conducted under a reducing atmosphere. Previous operando XANES and EPR studies have illustrated that the Cu(II) species in Cu/CeO₂ catalyst changed to Cu(I) under CO with a decrease of the Cu-O coordination as observed by EXAFS spectra, while the reverse was happening when CO* was removed by the inert gases He or N₂ gases only instead of by an oxidant as oxygen (Ref. 1-3). For CO adsorption on the Cu/CeO₂ catalyst, the phenomenon can be interpreted as hemilability that CO adsorption

opened the Cu-O coordination site to produce Cu(I) with a stronger binding ability. In other words, CO binds strongly on the adsorbed Cu/CeO₂ and promotes the reduction of Cu(II) species to Cu(I). Thus, there occurs structural changes and only Cu⁺ signal can be detected in CO-probe molecule FTIR measurements.

Reference:

1. Z. Chen, Z. Y. Liu, X. Xu. *Nat. Commun.*, **2023**, 14, 2512.
2. L. Q. Kang, B. L. Wang, Q. M. Bing, *et al. Nat. Commun.*, **2020**, 11, 4008.
3. F. Maurer, J. Jelic, J. J. Wang, *et al. Nat. Catal.*, **2020**, 3, 824-833.

This has been revised in manuscript: Line 8, Page 8.

Comments 4: The author should explain why the introduction of Cu element into parent CeO₂ reduce its surface Ce³⁺ concentration.

Reply 4: Based on earlier finds (Ref. 1), the replacement of one Ce³⁺ next to an oxygen vacancy (V_O) by a Cu²⁺ normally accompanies a phenomenon that the other Ce³⁺ would be readily converted to Ce⁴⁺ for the charge balance from the calculations and structural characterization results, and thus the increase of Cu²⁺ doping level will lead to a decrease of Ce³⁺/Ce⁴⁺ ratio from the original undoped CeO₂.

Reference:

1. Y. F. Wang, Z. Chen, P. Han, *et al. ACS Catal.* **2018**, 8, 7113-7119.

Line 14-19, Page 6 in the revised main article.

Comments 5: Is the decrease of HORAC activity and the increase of POD, OXD, CAT, and SOD-like activities of Cu-CeO₂ due to the change in the valence state of Ce, Cu acting as a monoatomic catalyst, or the interaction between the two? Is there any data to support this?

Reply 5: Thank you for your rigorous consideration. We have supplemented the POD-like activity test and ROS generation experiment of the physical mixed system of free Cu²⁺ ions/CuO with CeO₂, as described in the manuscript: "In addition, we investigated the catalytic effect of the physical mixed system of free Cu²⁺ ions/CuO nanozyme with CeO₂ nanozyme. As shown in Supplementary Fig. 17, the initial rate of POD-like reaction of the two physical mixed systems was similar to that of CeO₂. As the reaction continued, the substrate conversion extent of the two groups at the end point of the reaction was similar to that of Cu-CeO₂. However, DCFH fluorescence detection (Supplementary Fig. 18) showed that for physical mixed systems, only

Cu²⁺+CeO₂ group showed significantly higher total ROS generation than CeO₂ group, but was still much lower than that of Cu-CeO₂ group, indicating that in the non-interacting Cu²⁺/CuO+CeO₂ physically mixed system, Cu could not inhibit the HORAC activity of CeO₂ effectively. Through the above results, we reaffirm the enhanced POD-like activity of Cu-CeO₂ nanozyme, and a prominent inhibitory effect of Cu single sites on the HORAC activity of CeO₂. Meanwhile, the interaction between Cu single sites and CeO₂ support also plays a key role in the regulation of catalytic activities. Therefore, by the aid of Cu single sites and its interaction with CeO₂ support, the effective regulation of the redox catalytic pathways of CeO₂ nanozyme was achieved.”

Supplementary Fig. 17. Time-dependent optical density change at 652 nm of TMB in POD reactions.

Supplementary Fig. 18. Fluorescent spectra of DCFH after 10 min reaction with H₂O₂ and different nanozymes.

This has been revised in manuscript: Line 8-16, Line 18-20, Page 12.

Supplementary Fig. 17, Supplementary Fig. 18 in Supplementary Information.

Comments 6: How about the pH of mouse skin wound infection? Why can the Cu-CeO₂ catalyze the ROS generation for antibacterial performance and ROS scavenging for antioxidant simultaneously? It seems two opposite performances. The authors should clearly explain these aspects.

Reply 6: We reviewed the relevant literature and further analysed the role of Cu-CeO₂ nanozyme in the healing process of infected wounds, as described in the manuscript: “According to previous literature, the pH value of normal intact skin tissue is weakly acidic (pH = 4-6), while in the case of wound infection, due to inflammatory stimulation, the local pH value tends to be weakly alkaline (pH ≈ 7.4) (Ref. 1). It was confirmed through *in vitro* experiments that Cu-CeO₂ catalytically decomposed H₂O₂ efficiently to generate ROS under pH = 7.4, achieving excellent antibacterial effects and good cyclic stability. These inspired us to further explore the therapeutic effect of Cu-CeO₂+H₂O₂ on *in vivo* infected skin wounds.” “Specifically, since Cu-CeO₂ generates a large amount of ·OH only when applied simultaneously with H₂O₂, and does not generate excess toxic ROS in the absence of H₂O₂, when H₂O₂ is completely consumed, the residual Cu-CeO₂ nanozyme may continue to scavenge endogenous ROS in the wound microenvironment. It is clear that within the scope of this study, Cu-CeO₂ + H₂O₂ system plays an important role in the early anti-infectious process.”

Reference:

1. T. Cui, J. Yu, C. F. Wang, et al. *Adv. Sci.* **2022**, 9, 2201254.

This has been discussed in manuscript: Line 13-19, Page 20; Line 5-9, Page 23.

For Reviewer #2:

Reviewer #2: The authors extensively revised the manuscript and performed additional experiments and calculations which strengthened their earlier findings. I support publication of this manuscript.

Thank you again for your precious time and the recognition of our work. Your previous comments have significantly help us to improve our manuscript and made us have a deeper understanding of the characterization and calculation of our system. Your rigorous comments will be of great help to our research in the future.

For Reviewer #3:

Reviewer #3: The authors have revised their manuscript according to the previous reviewers' comments, and the quality of the manuscript has been improved. I believe now the manuscript can be considered for publication in the current form.

Thank you very much for your kind and positive comments. We are truly grateful for your

thoughtful assessment of our manuscript and the encouraging feedback you have provided. Your constructive suggestions were invaluable and motivated us to continue expanding our research in this field.

Reviewers' Comments:

Reviewer #1:

Remarks to the Author:

The authors have fully addressed my concerns. I think it can be accepted now.